# Anticipatory postural control emerges from a predictive and optimized strategy for movement preparation
Tetsuro Funato [1] ✉, Miho Ogawa[1], Akira Konosu[1] & Dai Yanagihara[2,3]

Humans shift their center of mass (COM) forward before predictable backward floor tilts. This anticipatory movement is accompanied by selective activation of the gastrocnemius muscle (GC), which produces backward torque at the ankle even as the body moves forward. The control mechanism underlying this coordination between forward COM motion and GC activation remains unclear. Here we investigate whether such behavior can be explained within a predictive optimal control framework. In experiments with auditory cues, anticipatory COM shift and GC activation were observed, representing a use of a backward-acting muscle during forward movement. We then developed a musculoskeletal simulation model governed by model predictive control (MPC), which reproduced these patterns. The simulation suggests the body shifts forward by leveraging gravity, while GC activation stabilizes the ankle and prevents collapse. These findings indicate that anticipatory postural responses may result from optimizing predicted future states using internal models that incorporate gravitational dynamics for efficient motor preparation.

When humans anticipate an incoming disturbance, they proactively initiate movements beforehand to mitigate its effects. Early studies reported that leg muscle activity precedes arm-raising movements[1], as a form of anticipatory postural adjustment (APA) to environmental changes or upcoming actions[2]. Similar anticipatory phenomena have been observed prior to various movements, such as arm raising[3,4], reaching[5,6], and gait initiation[7,8], suggesting that postural adjustments provide essential functions that enable humans and animals to perform movements smoothly. Conversely, when predictive postural control is impaired by neurological disorders, such as cerebellar disease[9,10] or Parkinson's disease[11], deficits in postural maintenance[9] and difficulties with gait initiation[11] occur, significantly impacting daily life. In other words, predictive postural control is a fundamental function underlying motor control in humans and animals, and understanding its mechanisms is crucial for developing effective strategies to restore motor function in patients with neurological disorders.

Discussions demonstrating adaptive responses to external disturbances while standing began from the perspective of adaptive adjustments in reflex systems. When a translational disturbance directed forward in the anterior–posterior direction is applied while standing, reflexive activity of the gastrocnemius muscle (GC) with a latency of approximately 120 ms (long latency response) increases, suppressing body fluctuations associated with the disturbance. In contrast, when a tilting disturbance in the toe-up direction is applied, which induces ankle dorsiflexion and results in a backward perturbation of the body, this reflexive activity of the GC decreases, altering its response to allow the disturbance to be turned aside[12]. Here, the cerebral cortex is believed to be involved in the adjustment of such predictive reflexes[13], because changes in cortical excitability are observed before predictable disturbances occur while standing[14,15], and transcranial magnetic stimulation of the primary motor cortex causes changes in leg postural responses with long latencies[16]. Additionally, in patients with cerebellar disease, the intensity of long-latency responses decreases[10,17,18], and in patients with Parkinson's disease, as severity increases, inhibitory control over reflex responses to external disturbances diminishes[19,20]. It is believed that circuits involving the cortex, cerebellum, and basal ganglia contribute to adaptive activities for maintaining posture based on prediction[13]. Then, what rules govern the adjustment of postural responses while standing in these central nervous systems?

In the generation of adaptive activities of human movement, optimal control has been suggested to be involved[21]. Cortical activity is thought to generate optimal control, which in turn modulates feedback and contributes to the generation of long-latency responses[21,22]. A study examining the relationship between postural control and optimal control by disturbing humans while standing[23] reported that human responses can be explained by optimization that minimizes muscle activity. Information about the postural state while standing is obtained through multiple sensors, such as visual, proprioceptive, and somatosensory[24], and it is believed that optimal

[1]Department of Mechanical Engineering and Intelligent Systems, The University of Electro-Communications, Chofu, Japan. [2]Department of Life Sciences, The University of Tokyo, Tokyo, Japan. [3]Cognition and Behavior Joint Research Laboratory, RIKEN Center for Brain Science, Wako, Japan. ✉e-mail: funato@uec.ac.jp

state estimation by a Kalman filter is used to integrate this sensory information with different characteristics to estimate the state[25–27].

In the control of quiet standing through optimal control and optimal state estimation, control inputs are generated based on the states up to the present. Predictive postural control while standing, on the other hand, can involve more active body movements, such as shifting the center of mass (COM) position before disturbance[28,29]. Because anticipatory COM movements occur before the disturbance, their generation requires predictive control mechanisms. However, the computational principles that generate such predictive control inputs remain unclear. The question addressed in the present paper is whether these anticipatory control inputs can be reproduced as an extension of existing control frameworks based on state estimation and optimal control.

In optimal control, state estimation and optimization are performed based on internal models of body dynamics. By estimating this state not only up to the present but also into the near future, and using this prediction for control, it seems possible to generate control inputs before the state changes. One such method is known as receding horizon control or model predictive control[30,31]. In model predictive control, state prediction for the near future and optimization for the predicted state are repeated at regular intervals to generate control inputs at each time step. The intermittent movements observed in slow finger movements have been explained as model predictive control involving repeated short-term optimal control[32,33], and a tendency for the prediction and optimization periods to be longer with increasing skill has been pointed out[34]. Intermittent control has also been pointed out as one of the main control principles explaining posture control in quiet standing[35–37], and thus, model predictive control, which consists of short-term optimization, is likely to be used in posture control.

In the present paper, we hypothesize that humans generate anticipatory postural control through model predictive control, and we verify this hypothesis through human experiments and dynamical simulations using a musculoskeletal model (Fig. 1). By repeatedly conducting experiments in which a tilt disturbance is applied to a standing human subject after a cue, we investigate the characteristics of predictive movements before the tilt. We then construct a system model consisting of a musculoskeletal model of the body and a control model using model predictive control. Through dynamical simulation, we will investigate the behavior of the body and muscles and compare it with that of humans. Through these studies, we approach the control principles that generate human predictive postural control.

## Results
### Subjects shift their COM forward after cue and before disturbance

Ten subjects were asked to stand upright, and after 20 s of standing, the floor was tilted by 4 degrees in 1 s (Fig. 1). After applying 30 tilting disturbances without a cue, 30 trials were conducted in which an auditory cue was given approximately 2 s before the tilt and the tilting disturbance was applied. Figure 2A shows the COM movements and muscle activity during the 5 s before and after the onset of tilting for 20 trials without a cue and 20 trials with a cue in a representative subject (results for all subjects are shown in Supplementary Fig. S1). In both conditions, the first 10 trials were excluded to focus on the movements after the subjects had sufficiently adapted to the task. Comparing the trials with and without cues in Fig. 2A, only in the trials with cues did the COM move forward before tilting after the cue sound was given. In addition, the COM position after tilting shows less change in the trials with a cue than in those without a cue. To investigate whether such preceding COM shifts prior to the tilt were observed in each subject, the changes in the COM during the cue period (from cue start: CS to floor tilt start: FS) for all subjects are shown in Fig. 2B. In Fig. 2B, the results without a cue are shown in blue, and the results with a cue are shown in red. Since there was no CS in the trial without a cue, the mean CS (with FS time set as time 0 s) of the trial with a cue for each subject was used as the CS for both conditions with and without a cue (the average CS (standard deviation) of the trial with a cue for all subjects was −2.19 (±0.12) s). Figure 2B shows that, compared to no cue (blue), 7 out of 10 subjects (red) had a significant forward shift in their COM when a cue was given ($p < 0.05$; see exact $p$-values, $t$-values, and degrees of freedom (df) in Supplementary Table S1A). In addition, in all other subjects, the COM shifted forward on average during the cue period compared to the no-cue trial.

To determine whether the COM shift before tilting arose primarily due to cue-based prediction or was largely influenced by reflex gain adjustment or habituation during the experiment, we divided the 30 trials for each condition into stages of 5 trials each. For each stage, we calculated the displacement of the COM during the cue period (similar to Fig. 2B), yielding the results shown in Fig. 3A. The $p$-values from testing the effects of the two factors (cue presence/absence and trial stage) using a two-way ANOVA are also shown in Fig. 3A ($F$-values and df are listed in Supplementary Table S2A). The significant effect of cue presence shown in Fig. 3A indicates that, even when accounting for the trial stage factor, a significant shift due to the cue occurred in the same 7 subjects as in Fig. 2B. Significant effects of trial stage were observed in 2 out of 10 subjects. However, when testing the effect of trial stage separately for cue-free and cue-present conditions using a one-way ANOVA (Supplementary Table S2B), significant differences were found only for subject 5 ($p = 0.03$) and subject 10 ($p = 0.01$) in the cue-present condition. Figure 3A shows that these subjects exhibited a large initial shift without subsequent gradual changes. These results suggest that the COM shift before the tilt in this experiment was primarily driven by cue-based prediction, rather than by reflex gain adjustment or habituation.

To show what joint movements caused the COM shift, we present the time series of the angles of the hip and ankle in Fig. 2C (see also the postures at the start of the cue (CS) and at the start of the floor tilt (FS) in Supplementary Fig. S2A). As with the COM, when changes during the cue period were examined (Fig. 2D), changes in the hip joint due to the cue were observed in only two subjects, whereas changes in the ankle joint due to the

## Experiment

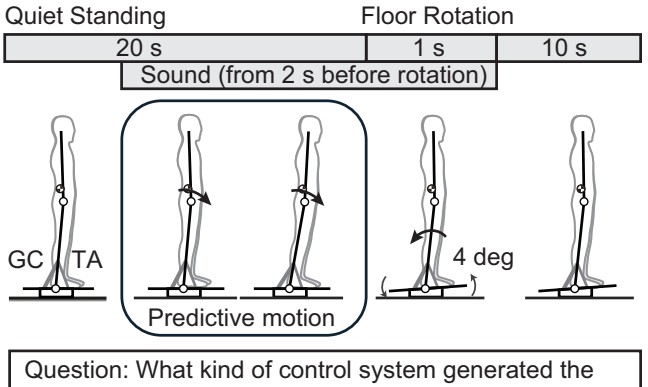

Quiet Standing | Floor Rotation
20 s | 1 s | 10 s
Sound (from 2 s before rotation)

GC TA

Predictive motion

4 deg

Question: What kind of control system generated the predictive motion?

## Assumption

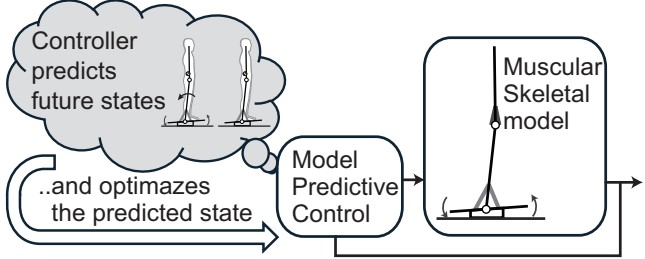

Controller predicts future states

..and optimazes the predicted state

Model Predictive Control

Muscular Skeletal model

**Fig. 1 | Overview of the experiment and system model.** To investigate posture control functions based on predictions, a tilt disturbance was applied to a platform on which a human subject was standing after receiving a cue. Assuming that the predicted movements generated before the disturbance are produced by model predictive control, this assumption was verified through dynamical simulation.

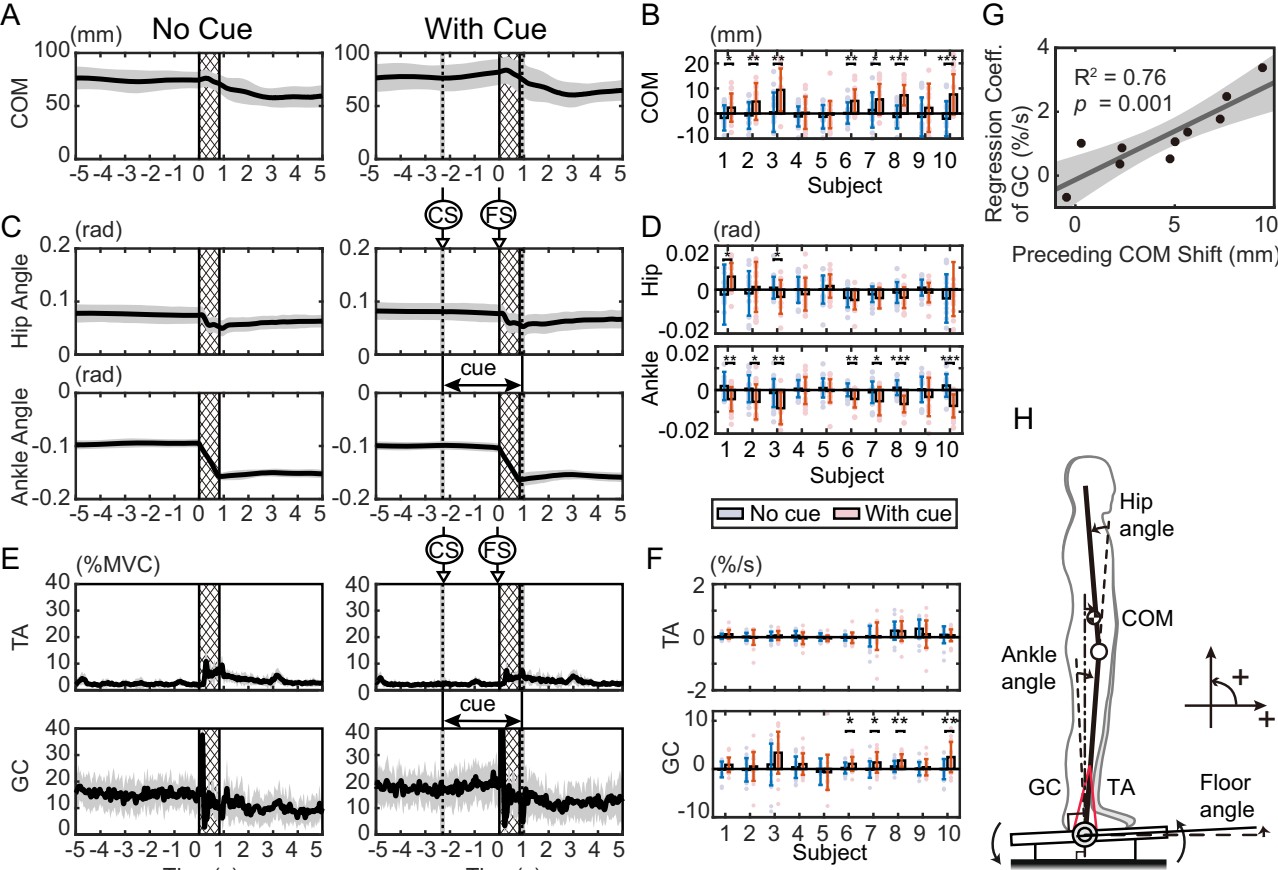

**Fig. 2 | Experimental results.** Time series of center of mass (COM) position (**A**), hip and ankle angles (**C**), and muscle activity (**E**) for 5 s before and after the start of the tilt. The anteroposterior position of the COM is shown with the ankle position as 0 (mm). In each panel, the trials without a cue and with a cue are shown separately. The results of 20 trials for one representative subject (excluding the first 10 trials in each condition) are shown as the mean (solid line) and standard deviation (SD; gray area) in each condition ($n = 20$ trials per condition). The slanted area represents the floor tilt period, and the dotted line (area behind the dotted line) in the trials with a cue indicates the mean (SD) of the start and stop times of the cue sound. TA tibialis anterior, GC gastrocnemius. Changes in COM position (**B**), hip and ankle joint angles (**D**), and muscle activity (**F**) from cue start (CS) to floor tilt start (FS) for each subject. For each subject, values are shown as the mean and SD across 20 separately performed trials in each condition ($n = 20$ trials per condition). In **B**, **D** the difference between the values at CS and FS is shown. In **F** the regression coefficient obtained from linear regression between CS and FS is shown. Blue bars indicate the no-cue condition, and red bars indicate the cue condition. The mean CS (with FS time set as time 0 s) of the trials with cue for each subject was used as the CS for both conditions. *: $p < 0.05$, **: $p < 0.01$, ***: $p < 0.001$. **G** Relationship between COM shift (**B**) and the regression coefficient for GC activity (**F**) during the cue period for each subject. Each point represents the average across cue trials for each subject. The line represents the fitted linear regression, and the gray area around the line represents the 95% confidence interval. **H** Definition of COM position, joint angles, and floor angle.

cue were observed in all seven subjects who showed a COM shift. This indicates that the COM shift during the cue period was mainly caused by movement of the ankle.

### Forward COM shift is accompanied by posterior-ankle muscle activation

To investigate what muscle activities are responsible for this shift in the COM, we present the activities of the anterior and posterior muscles of the ankle (TA: tibialis anterior and GC: gastrocnemius) in Fig. 2E (results for all subjects are shown in Supplementary Fig. S3). The figure shows that TA activity was smaller than GC activity, with only slight activity observed after the start of the tilting. This indicates that predictive postural control is mainly performed by the GC. Time series analysis revealed a gradual increase in GC activity during the cue period. To quantify individual muscle activity changes during the cue period, we applied linear regression and plotted the resulting coefficients in Fig. 2F. In the figure, positive values represent an increase in activity during the cue period, while negative values represent a decrease in activity. Blue represents the results of trials without cues, and red represents the results of trials with cues. The figure shows that there was almost no TA activity in either condition, but in GC, the

coefficient was significantly higher in trials with cues in four subjects ($p < 0.05$, t-test; see exact $p$-values, $t$-values, and df in Table S1B), all of whom showed significant differences in Fig. 2B. In addition, the average GC coefficient was higher in trials with cues in all subjects except the fifth subject.

Similar to the COM shift, muscle activity changes during the cue period were evaluated across trial stages, yielding the results shown in Fig. 3B. Figure 3B presents the linear regression coefficient of GC activity during the cue period (similar to Fig. 2F), divided into five trial stages. The $p$-values from testing the effects of cue presence and trial stage using a two-way ANOVA are shown in Fig. 3B ($F$-values and df are listed in Supplementary Table S3A). Results for the TA are presented in Supplementary Fig. S4 and Supplementary Table S4. However, TA activity during the cue period averaged less than 6% across all trial stages, and ANOVA revealed significant effects of both trial stage and cue presence in at most one subject. Figure 3B shows that subjects 3, 6, 8, and 10 exhibited changes depending on cue presence. Among these, subjects 6, 8, and 10 also showed consistent changes in Fig. 2F, while subject 3 was the same individual who exhibited COM shift changes in Fig. 2B. Significant effects of trial stage were observed in subjects 2 and 10. When comparing stage-specific differences using one-

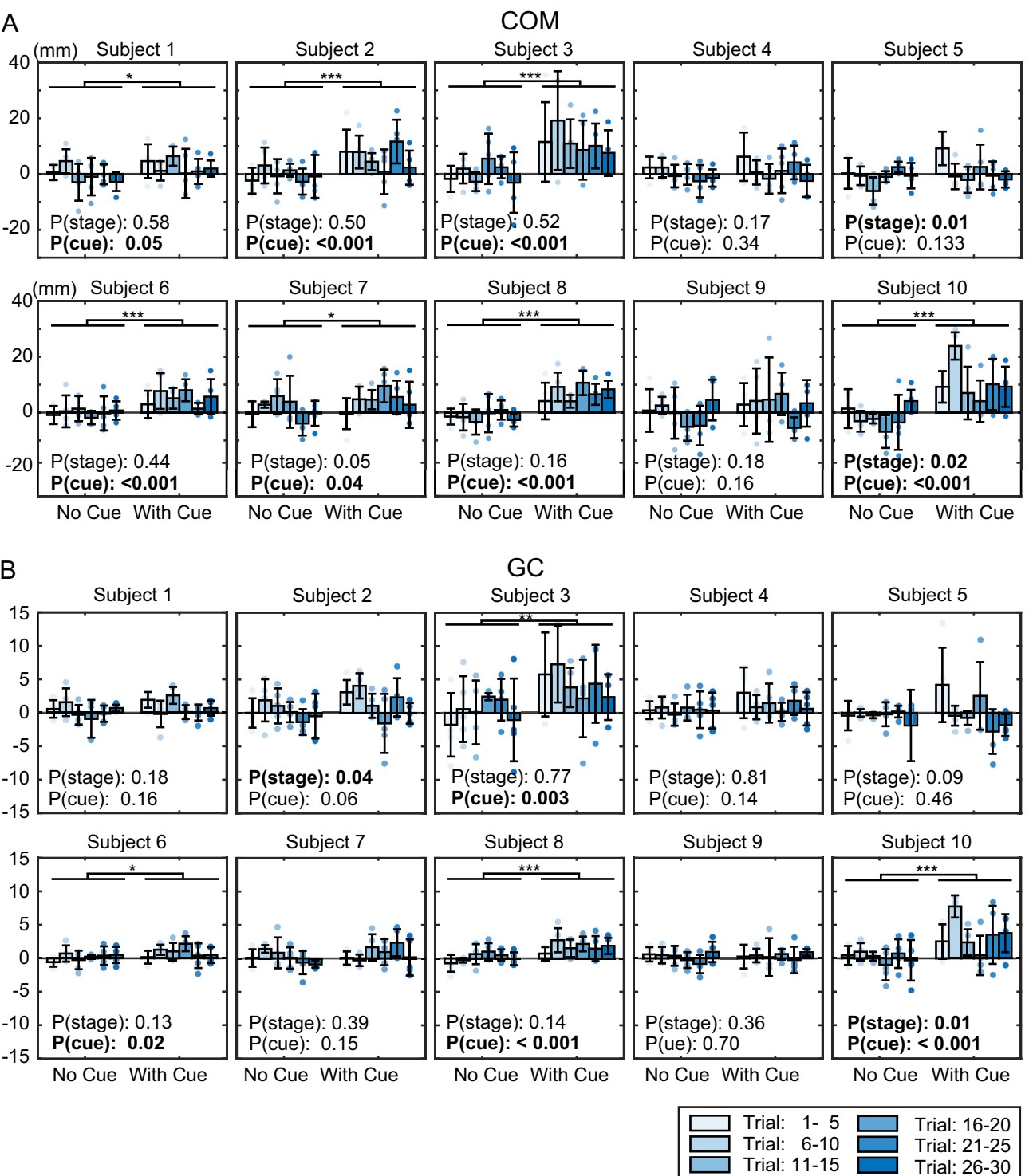

**Fig. 3 | Effect of trial stage on center of mass (COM) displacement and gastro-cnemius (GC) activity changes. A** COM displacement and **B** GC activity changes during the cue start (CS)-floor tilt start (FS) period. Each condition (no cue and cue) consists of 30 trials divided into 6 stages of 5 trials each. For each subject, values in each stage are shown as the mean and SD across the trials within that stage ($n = 5$ trials per stage). **A**, **B** Correspond to the analyses shown in Fig. 2B, F, respectively, separated by trial stage. The *p*-values in the figure represent the results of a two-way ANOVA for trial stage and cue presence. Significant differences due to the cue are indicated by * in the figure. *: $p < 0.05$, **: $p < 0.01$, and ***: $p < 0.001$.

way ANOVA separately for the cue-free and cue-present conditions (Supplementary Table S3B), no significant differences were observed in any subjects under the cue-free condition. In contrast, in the cue-present condition, significant differences were observed in subjects 1, 2, 5, and 10. Linear regression analysis of the stage-specific values for these subjects (Supplementary Table S3C) revealed negative regression coefficients in all cases,

with *p*-values less than 0.05 for subjects 2 and 5. These results, along with the absence of stage-specific changes under the cue-free condition, suggest that cue-induced changes are the primary contributors to GC activity, similar to the COM results. Furthermore, the stage-specific differences observed in the cue-present condition appeared to reflect reduced GC activity changes over time, indicating that responses to cues were likely adjusted across trials.

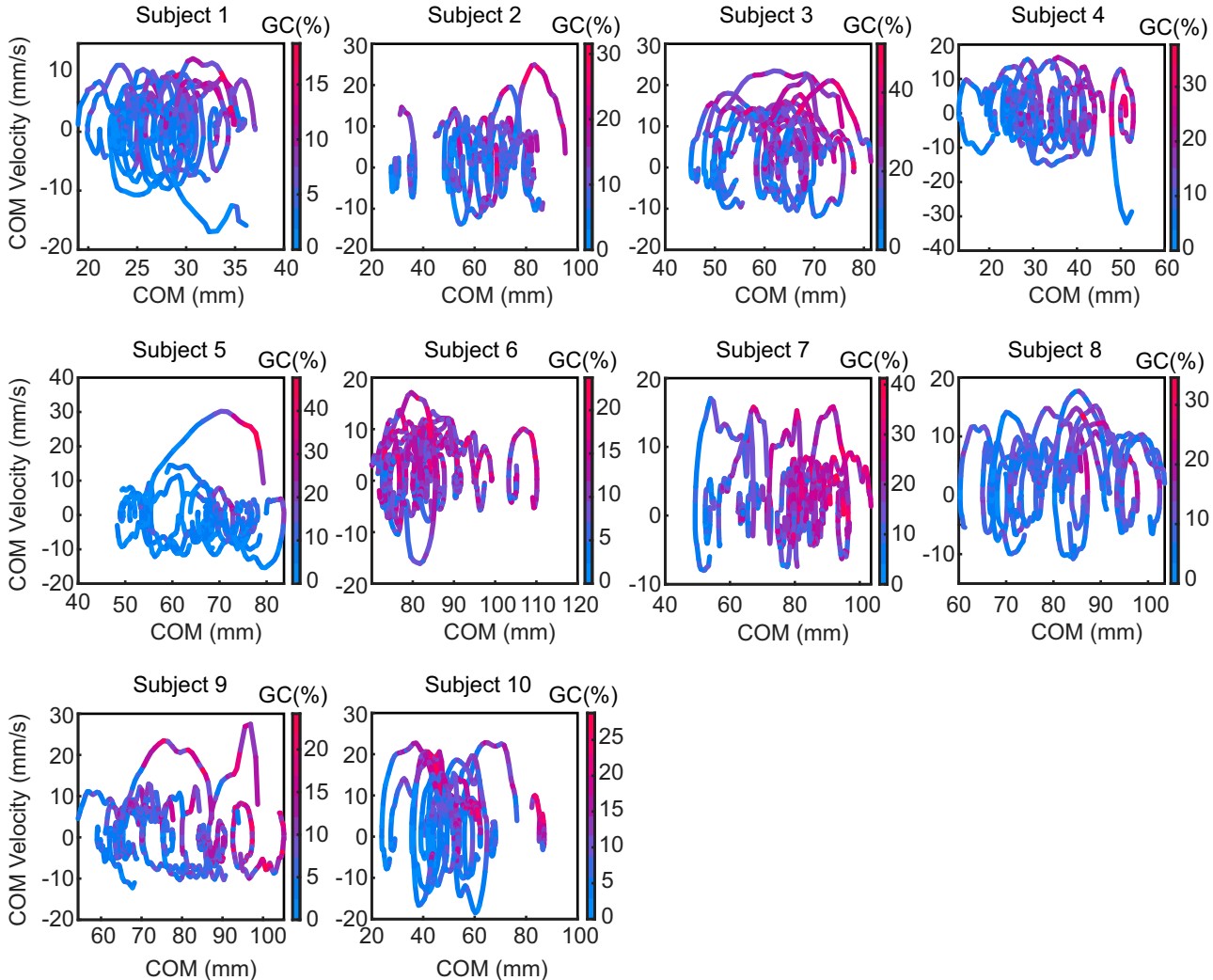

**Fig. 4 | Phase portrait of the center of mass (COM) and gastrocnemius (GC) activity during the cue start (CS)-floor start (FS) period.** The curves represent the relationship between COM position and COM velocity from CS to FS. The color of the trajectories indicates the magnitude of GC activity at each state. Each panel shows data from each subject, with trajectories from 20 cue trials overlaid.

## What control mechanism can explain the forward shift of the COM accompanied by posterior GC activity?

When comparing the direction of the COM (Fig. 2A) and muscle activity (Fig. 2E), it is noticeable that an increase in GC activity occurs during forward movement of the COM. To examine how GC activity relates to COM shift, we plotted the relationship between the magnitude of COM shift during the cue period and the rate of change in GC activity over the same period (Fig. 2G). The results showed that GC activity increased as the COM shift increased (linear regression: $R^2 = 0.76$, $p = 0.001$). The GC muscle is located posterior to the ankle joint, and its activation generates a posterior ankle torque. This figure suggests that as the COM moves further forward, GC activity increases to counteract this displacement and maintain postural balance.

To further investigate the COM states in which GC is activated, we plotted the phase portrait of the COM[38,39] (the relationship between COM position and COM velocity) during the cue period in Fig. 4, with GC activity at each state represented by color. As shown in the figure, the COM-COM velocity relationship forms a cyclic state pattern similar to that of quiet standing, and the COM exhibits a state transition in which it moves forward while preserving the cyclic shape. Furthermore, GC activity is particularly pronounced in the upper-right region of each cycle, corresponding to states in which the COM is located anteriorly, and the COM velocity is positive.

This indicates that GC activity repeatedly switches its intensity depending on the state of the COM cycle. In other words, the COM limit-cycle dynamics and intermittent muscle activation previously reported during quiet standing[38,40] are also suggested to be utilized during the predictive period.

The relationship between COM motion and GC activity described above is not inconsistent with the role of GC activity in maintaining postural stability. However, when focusing on the fact that GC can generate only posterior ankle torque while the COM is being shifted forward, the direction of the generated torque is opposite to the direction of the intended COM shift. This inversion between the direction of control and the direction of muscle-generated torque suggests that forces other than muscle-generated torque contribute to moving the COM forward.

A similar relationship between the direction of COM motion and ankle torque has been observed in previous studies of quiet standing[41,42] and has been interpreted in terms of the relationship between the COM and the center of pressure (COP)[38]. These studies have shown that forward sway during quiet standing can arise through a "drop-and-catch" mechanism[42,43]. In this mechanism, relaxation of the plantarflexor muscles causes the COP to move posterior to the COM, resulting in forward acceleration of the COM (drop phase). As the COM moves forward, the plantarflexors are

**Fig. 5 | System model used in the simulation.** The system model consists of a body model based on a musculoskeletal model and a control model based on model predictive control, and is defined as a two-dimensional model that moves only in the sagittal plane. The body is composed of a two-link inverted pendulum and a tilting floor and is driven by four muscles: anterior and posterior ankle muscles (tibialis anterior TA; gastrocnemius, GC) and anterior and posterior hip muscles (iliopsoas, IL; gluteus maximus, GM). The activity levels of the muscles are determined by model predictive control to generate the torque for movement.

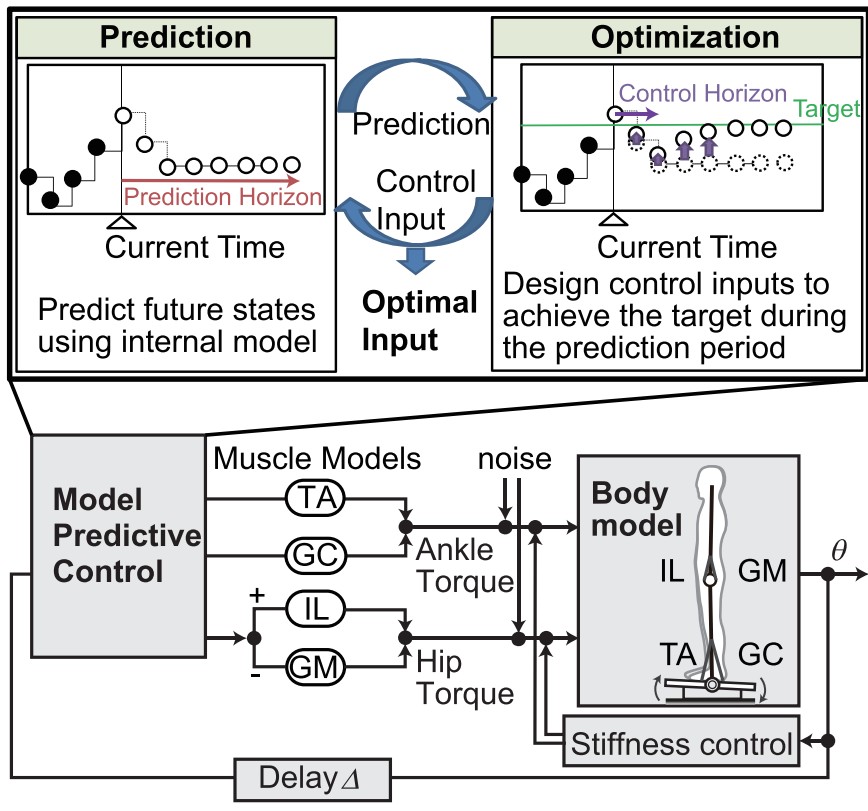

subsequently activated, shifting the COP anterior to the COM. This reduces the forward motion of the COM and stabilizes it at a new position (catch phase).

These studies suggest that biomechanical effects arising from relaxation of the plantarflexor muscles, particularly the action of gravity, may be utilized in postural control. During upright standing, gravity is continuously acting on the body, and as posture becomes more forward-leaning, gravity generates a stronger forward-directed torque. By simultaneously utilizing the forward torque generated by gravity and the posterior torque generated by GC activity, it becomes possible to shift the COM forward while maintaining stability. However, implementing this strategy requires estimating the gravitational torque and generating GC activity based on the difference between the gravitational torque and the desired net torque. Because the magnitude of gravitational torque varies continuously with posture, estimating this torque necessitates an internal model that represents the relationship between posture and torque. In other words, by predicting body states that include gravitational torque based on an internal model and generating optimal control inputs accordingly, it may be possible to reproduce a control strategy that explicitly exploits gravity in the generation of postural control inputs.

In the following sections, we will examine the mechanism that generates the preceding COM shift using a mathematical model. In particular, we will investigate whether model predictive control, which generates control inputs through state prediction and optimization of the predicted state, can produce the preceding behavior observed in humans. In examining the control mechanism, we will focus on the muscle activity during the COM shift as well as the generation of the preceding COM shift.

**Simulation using predictive optimization reproduces the COM shift and GC activation**

A musculoskeletal model consisting of a two-link skeletal system with joints at the ankle and hip, with muscles arranged anteriorly and posteriorly at the ankle and hip, was constructed, and control inputs were applied using model predictive control (Fig. 5). The ankle is fixed to a tilting floor, and

disturbance is introduced by tilting the floor at a rate of 4°/s, similar to human experiments. Model predictive control predicts future states based on an internal model and generates control inputs to optimize the predicted future states. In this study, we assumed that the internal model had sufficiently learned the floor tilt disturbance and used the same equations of motion of the system model as the internal model. The initial angles of the ankle and hip in the model were set to the state at the CS time in the experiment with a cue, and the COM position at that time (average of all subjects: 61.5 (±21.1) mm, rounded to 61 mm) was used as the target COM position for the optimization evaluation function. Using this mathematical model, a simulation was conducted for 20 s under conditions where external disturbances were input at 15–16 s, and the external disturbance information was reflected in the predictive control from 2 s before the external disturbance (13 s).

The simulation results are shown in Fig. 6. Figure 6A shows that the COM moves forward from around 2 s before the start of the tilt (after the cue). In other words, when sequential optimization was performed to optimize the future state using model predictive control, the COM shifted forward from the target position before the tilt (Fig. 6C, D shows the change of COM, hip, and ankle in the CS-FS period. COM: $t = -26.3$, df = 38, $p < 0.001$; hip: $t = 2.64$, df = 38, $p = 0.01$; ankle: $t = 8.91$, df = 38, $p < 0.001$). Next, we focus on muscle activity (Fig. 6E). Figure 6E shows that, similar to the muscle activity patterns observed in the experiment, TA remains almost inactive while GC increases as the COM shifts forward.

To investigate the relationship between COM shift and GC activity, we plotted the relationship between COM shift and changes in GC activity for each participant between CS and FS (Fig. 6G). The results revealed a relationship similar to that observed experimentally (Fig. 2G), in which greater COM displacement was associated with larger changes in GC activity (linear regression: $R^2 = 0.71$, $p < 0.001$). Furthermore, as in Fig. 4, we plotted the phase portrait of COM and represented GC activity using color, yielding the pattern shown in Fig. 6H. The trajectory initially exhibited a limit-cycle-like pattern during the early phase of the predictive period (left side), after which it shifted along a semicircular path. On the right side, where the COM was

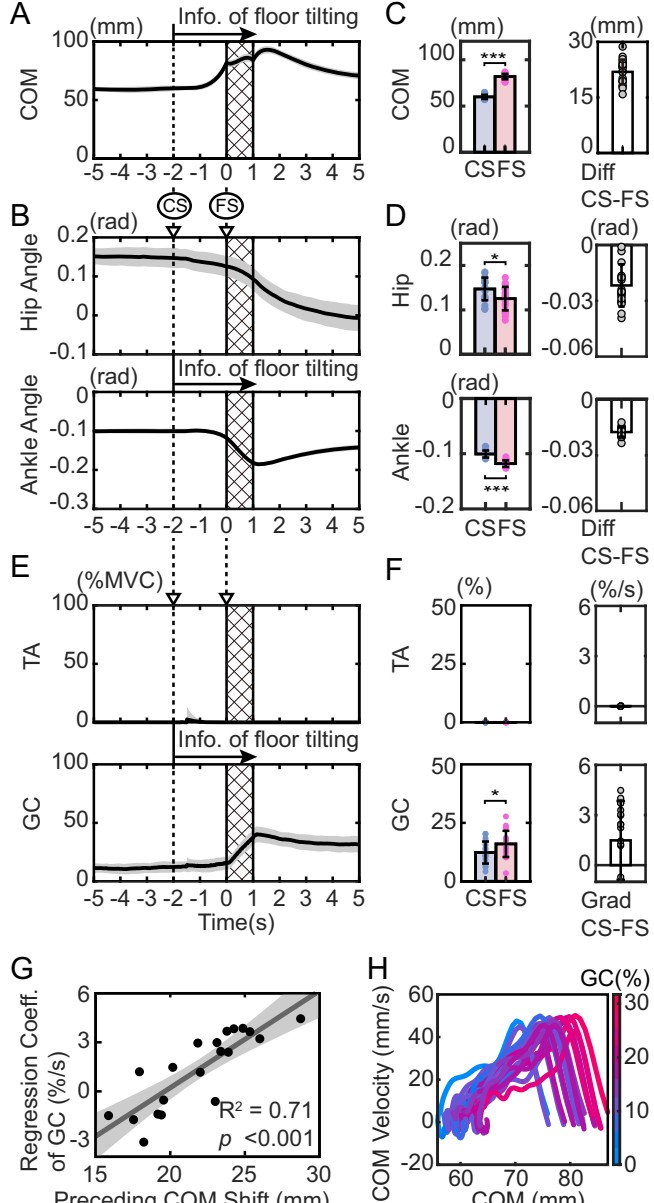

positioned anteriorly, GC activity increased, accompanied by a reduction in COM velocity. These results demonstrate trajectory and activity patterns similar to those observed in the experimental data (Fig. 4).

Based on these results, we examine what occurs between the COM and GC during the CS-FS period. Here, we focus on the COM movement (Fig. 6A) and ankle and hip joint angles (Fig. 6B) at the beginning of the cue (CS). At CS, the ankle angle is negative (anterior tilt), the hip angle is positive (posterior tilt), and the COM is positive (forward) (see also the postures at CS in the simulation in Supplementary Fig. S2B). In this posture, when the activity of both the TA and GC muscles decreases simultaneously, the entire body folds down, the body leans forward around the ankle, and the COM shifts forward. Since this forward tilt around the ankle must be prevented to avoid falling, it is necessary to increase GC activity to suppress it. In other words, the simulation (Fig. 6) also reproduced the movement strategy observed in the experiment (Fig. 2), in which the body leans forward around the ankle in accordance with gravity, shifts the COM anteriorly, and gradually increases GC activity. These findings further suggest that the efficient movement strategy obtained through optimization may leverage gravity-induced COM shift to minimize active muscle use.

## COM shift and GC activity reflect energy-efficient control strategy

To verify whether such gravity-based motion strategies actually arise in relation to the efficiency of control inputs, we performed simulations by varying the weight of control inputs in the evaluation function (Fig. 7). The weight of control inputs in Fig. 6 was set to 0.01, and in Fig. 7, it was set to 20 values ranging from 0.001 to 0.1 in logarithmic intervals. From Fig. 7A, B, it can be seen that as the input weight increases (emphasizing energy efficiency), both the hip and ankle angles approach 0, gradually moving toward a vertical standing posture. Muscle activity (Fig. 7C, D) shows that when the input weight is small, both TA and GC are used to maintain the posture. However, as the input weight increases, TA activity decreases, and at a weight of approximately 0.01, only GC is active at both CS and FS timings. Furthermore, as the input weight increases further, GC activity increases at both CS and FS timings. At this time, gluteus maximus (GM) activity decreases, and control is performed using the hip rather than the ankles. These results indicate that when the weights of the inputs are small (when the efficiency of the control inputs is not important), the control strategy of active control using muscle activity is used, whereas as the weights increase (when the efficiency of the control inputs becomes important), the control strategy of using gravity and coordination by the GC appears. The muscle activity observed in human experiments (Fig. 2E) was the latter, suggesting that control by GC resulted from efficient use of control inputs.

## Predicted disturbance magnitude determines predictive postural adjustments

We next asked what factors determine the amount of forward COM shift during the prediction period. In the simulation, we systematically varied the target position of the COM (Fig. 8A–C). Note that the target position in Fig. 6 was 61 mm forward of the ankle, whereas in Fig. 8A–C it was varied in 10-mm increments from −50 mm to +100 mm. Here, in the simulation, the target position and the initial position were always set to be identical in order to reduce the influence of transient movements from the initial position to the target position and to focus on movements based on prediction. Figure 8A shows that the body moved forward by approximately the same amount under all conditions, regardless of the target position. As shown in Fig. 8B, TA activity increased as the target position moved backward, whereas GC activity increased as it moved forward. This modulation of TA and GC activity, corresponding to postural differences, reflects the extent to which gravity can be utilized to shift the COM. Based on these results, we examined the factors that determine the forward displacement of the COM during the prediction period. In particular, we considered whether the COM shifts toward a specific optimal position, or whether the amount of movement remains constant irrespective of the reference. The simulation results with varied COM targets (Fig. 8) revealed a consistent forward shift of the

**Fig. 6 | Results of the dynamical simulation.** Time series of COM position (**A**), hip and ankle angles (**B**), and muscle activity (**E**) for 5 s before and after the start of the tilting. The results of 20 trials with different random noise realizations are shown as the mean (solid line) and standard deviation (gray area) ($n = 20$ independent simulation trials). The slanted area represents the period of floor tilt, and the dotted line indicates the time at which the tilt became predictable (CS). TA tibialis anterior, GC gastrocnemius. COM position (**C**), hip and ankle angles (**D**), and muscle activity (**F**) at cue start (CS) and floor tilt start (FS). In **C**, **D** the panels labeled CS and FS show COM position and hip and ankle joint angles at cue start and floor tilt start, respectively. The panels labeled Diff CS-FS show the differences in COM position and hip and ankle joint angles between FS and CS. In **F** the panels labeled CS and FS show TA and GC activity at cue start and floor tilt start, respectively. The panel labeled Grad CS-FS shows the regression coefficients obtained from linear regression over the period from CS to FS. All values are presented as the mean and SD ($n = 20$ independent simulation trials). *: $p < 0.05$; ***: $p < 0.001$. **G** Relationship between COM shift (**C**) and the regression coefficient for GC activity (**F**) during the cue period. Each point represents one trial. The line represents the fitted linear regression, and the gray area around the line represents the 95% confidence interval. **H** Phase portrait of the COM and GC activity during the CS-FS period. The color of the trajectories indicates the magnitude of GC activity at each state. The results of 20 trials are overlaid.

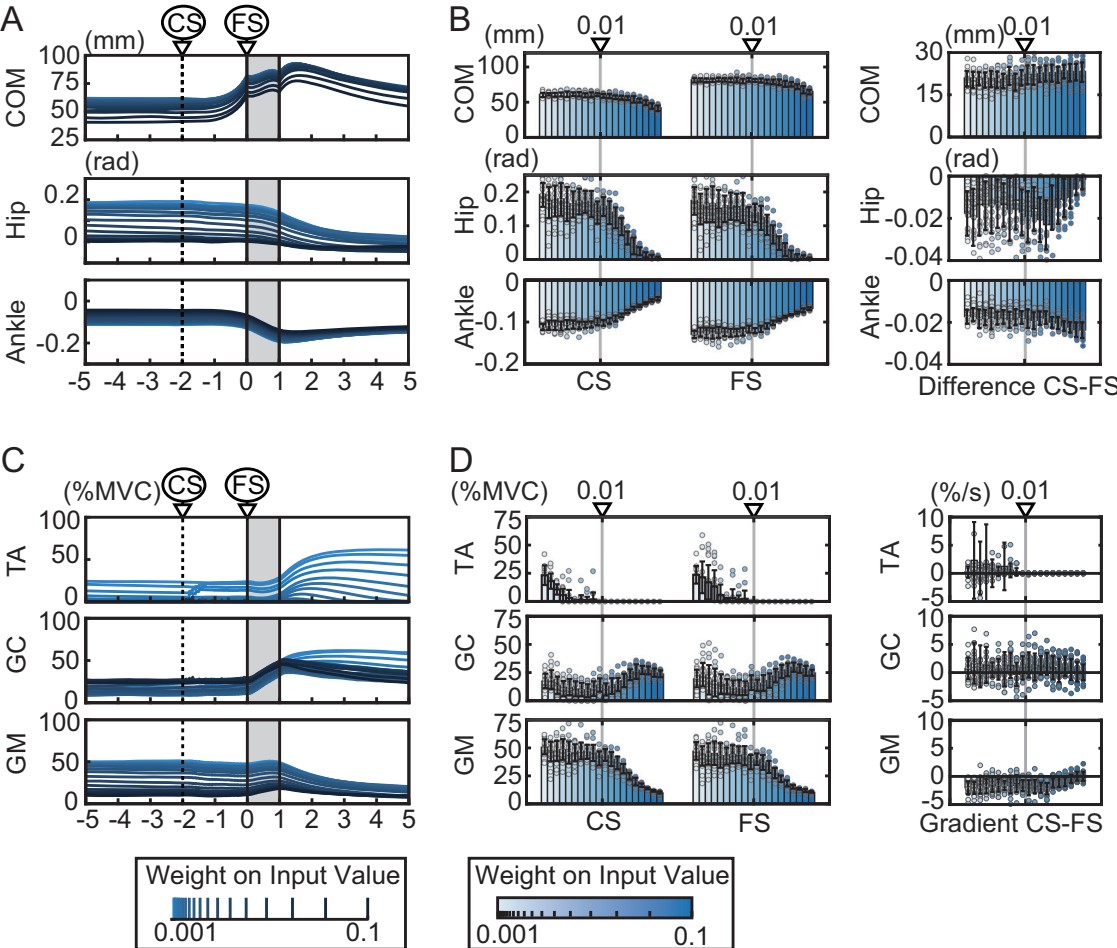

**Fig. 7 | Effect of input weights in model predictive control on predictive behavior in the simulation.** The results are from simulations in which the input weights in the evaluation function of model predictive control were varied from 0.01 to 0.1 in logarithmic intervals. Each data point represents the mean of 20 trials executed with different random noise. **A** Results for the COM position, hip joint angle, and ankle joint angle. CS indicates the time at which the tilt became predictable, and FS indicates the onset of floor tilt. **B** The panels labeled CS and FS show COM position and hip and ankle joint angles at cue start and floor tilt start, respectively. The panels labeled difference CS-FS show the differences in COM position and hip and ankle joint angles between FS and CS. All values are presented as the mean and SD ($n = 20$ independent simulation trials). **C** Results for muscle activity. TA tibialis anterior, GC gastrocnemius, GM gluteus maximus. **D** The panels labeled CS and FS show TA, GC, and GM activity at cue start and floor tilt start, respectively. The panel labeled gradient CS-FS shows the regression coefficients obtained from linear regression between CS and FS. All values are presented as the mean and SD ($n = 20$ independent simulation trials).

COM, irrespective of the reference position. We then examined the extent of movement during the prediction period following the cue. Changes in COM, hip, and ankle angles during the cue period (between CS and FS) are shown in Fig. 8C. The COM shifted forward by approximately 20–30 mm across all target positions, with no significant effect of target position (one-way ANOVA: $F = 1.45$, df = 15, $p = 0.123$). In contrast, hip and ankle angles significantly varied depending on the target position (hip: $F = 10.26$, df = 15, $p < 0.001$; ankle: $F = 9.08$, df = 15, $p < 0.001$; one-way ANOVA). As shown in the figure, the ankle moved forward as the COM target position increased, while the hip moved backward, resulting in compensatory joint movements that appeared to offset the overall COM displacement. These findings suggest that the preceding forward shift of the COM in response to floor tilt does not cause subjects to adopt a specific posture, but rather induces a consistent forward shift of approximately 20–30 mm, regardless of their initial posture.

How, then, is this amount of COM shift determined? To address this, we conducted simulations varying the magnitude of the floor tilt disturbance. Figure 7D–F show the results of simulations in which the floor tilt angle was incrementally increased from 1 to 8 degrees. Figure 8D shows that while the COM position remained nearly constant at CS, it shifted progressively forward at FS as the tilt angle increased. As shown in Fig. 8F, the amount of COM shift during the cue period (between CS and FS), as well as

changes in hip and ankle angles, significantly varied with tilt magnitude (one-way ANOVA: COM: $F = 380.5$, df = 7, $p < 0.001$; hip: $F = 7.79$, df = 7, $p < 0.001$; ankle: $F = 240.0$, df = 7, $p < 0.001$). These results suggest that the preceding COM shift in anticipation of the floor disturbance is determined by the predicted magnitude of that disturbance, rather than by a movement toward a specific postural configuration.

### Extension of the model to reproduce reflexive activity during floor tilt

The proposed model was designed to reproduce the forward COM shift and GC activity during the predictive period preceding floor tilt, and showed that these behaviors can be reproduced by model predictive control. However, when muscle activity during the floor tilt itself was examined, differences between the experimental and simulation results became evident. In the experiments, GC exhibited a transient, impulse-like activation following floor tilt onset, followed by TA activation (Fig. 2E; similar responses were observed in other participants, Supplementary Fig. S3). In contrast, the simulation showed a gradual increase in GC activity during the same period, and no TA activation was observed (Fig. 6E).

To characterize the experimentally observed impulse-like GC activity, we quantified the timing of GC activation during the tilt period. The first

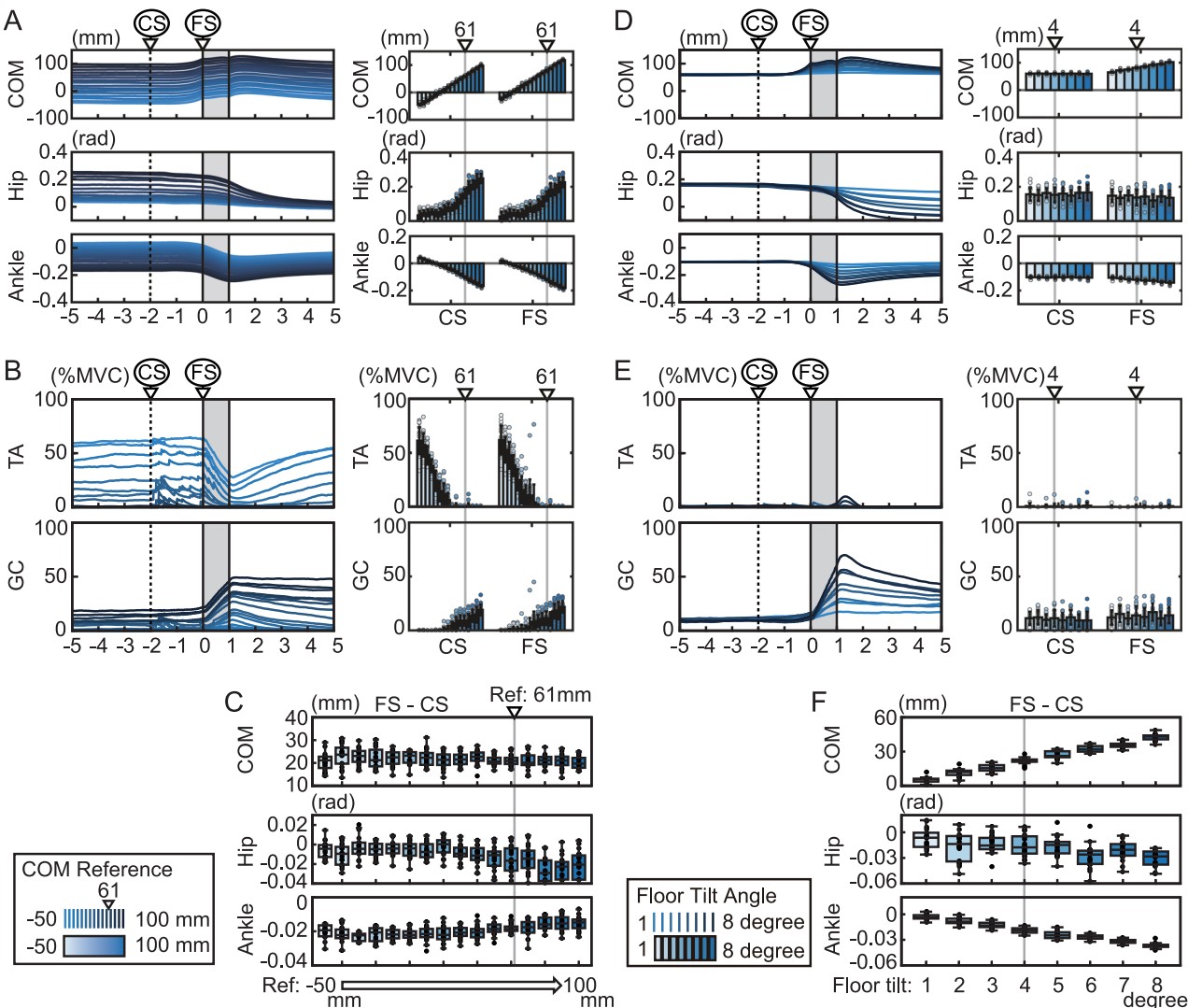

**Fig. 8 | Effect of the center of mass (COM) reference position and maximum floor tilt on predictive behavior in the simulation. A–C** Results of simulations in which the COM reference position for model predictive control was varied at equal intervals from −50 mm to +100 mm. **D–F** Results of simulations in which the maximum tilt angle of the floor was varied at equal intervals from 1 degree to 8 degrees. Each result shows the average of 20 trials performed with different random noise. **A, D** Results for COM position, hip joint angle, and ankle joint angle. CS indicates the time at which the tilt became predictable, and FS indicates the onset of floor tilt. **B, E** Results for muscle activity. TA tibialis anterior, GC gastrocnemius. **C, F** Changes in COM position, hip joint angle, and ankle joint angle between CS and FS. Values in the bar charts are presented as the mean and SD ($n = 20$ independent simulation trials).

peak of GC activity occurred at $83.0 \pm 52.6$ ms in the no-cue condition and $85.0 \pm 50.7$ ms in the cue condition after tilt onset (see Supplementary Table S5 for peak latencies of individual subjects). These latencies correspond to the early component of the long-latency response (LLR)[12]. No significant effect of cue condition on peak latency was observed (two-way ANOVA with factors subjects and cue condition: subjects, $F = 1.17$, df = 9, $p = 0.31$; cue condition, $F = 0.161$, df = 1, $p = 0.69$).

Based on these observations, we extended the control input by adding muscle activity corresponding to the LLR, in addition to the existing model predictive control-based input. Specifically, GC activation was applied during a 100-ms window following floor tilt onset. With this extension, the simulation reproduced an impulse-like activation of the GC after tilt onset, followed by TA activity (Fig. 9E). Consistent with the experimental results (Fig. 2A), the COM motion decreased during the tilt phase (Fig. 9A), and a rapid decrease in hip angle following tilt onset was also reproduced (Fig. 9B). The transient increase in COM observed after tilt termination in Fig. 9A was not present in Fig. 2A (subject 7), but similar behavior was observed in other participants (subjects 1, 2, 4, and 9; Supplementary Fig. S1).

## Discussion

Before environmental changes occur, humans often generate anticipatory movements that involve active shifts of the COM. To explain such active movements before environmental changes, we hypothesized that these anticipatory COM movements arise from predictive optimization of future states based on internal models. We investigated this mechanism through human experiments and dynamical simulations. When healthy subjects were given a cue and the floor was tilted, a forward COM shift was observed, accompanied by increased GC muscle activity. We performed a dynamical simulation using a control system based on model predictive control and compared the results with the experimental data. The results showed that the behavior observed in the experiment, namely the forward shift of the COM and the activity of the GC after the cue, was reproduced. In dynamical simulations, emphasizing control efficiency during optimization resulted in control by GC, as observed in experiments. This suggests that the activity of the GC during the preceding forward shift of the COM is a result of optimizing control input efficiency. When shifting the COM forward, rather than actively controlling it through TA, relaxing the muscles allows the body to tilt forward due to gravity, thereby shifting the COM forward. The

**Fig. 9 | Results of dynamical simulations using a model that incorporates responses to floor tilt onset.** Time series of COM position (**A**), hip and ankle angles (**B**), and muscle activity (**E**) for 5 s before and after the start of the tilting. The results of 20 trials with different random noise realizations are shown as the mean (solid line) and standard deviation (gray area) (*n* = 20 independent simulation trials). The slanted area represents the period of floor tilt, and the dotted line indicates the time at which the tilt became predictable (CS). TA tibialis anterior, GC gastrocnemius. COM position (**C**), hip and ankle angles (**D**), and muscle activity (**F**) at cue start (CS) and floor tilt start (FS). In **C, D** the panels labeled CS and FS show COM position and hip and ankle joint angles at cue start and floor tilt start, respectively. The panels labeled Diff CS-FS show the differences in COM position and hip and ankle joint angles between FS and CS. In **F** the panels labeled CS and FS show TA and GC activity at cue start and floor tilt start, respectively. The panel labeled Grad CS-FS shows the regression coefficients obtained from linear regression over the period from CS to FS. All values are presented as the mean and SD (*n* = 20 independent simulation trials). *: *p* < 0.05; ***: *p* < 0.001.

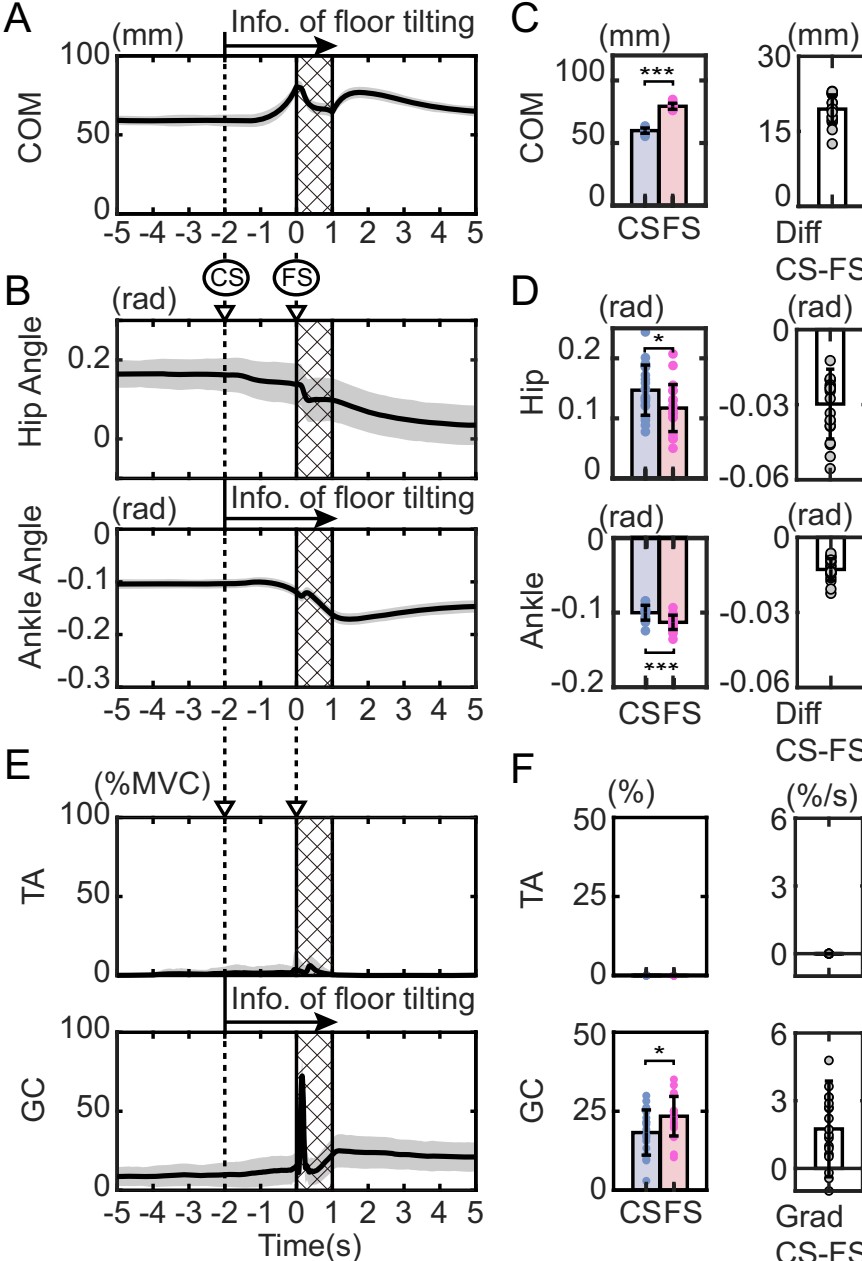

forward shift can then be modulated by GC activity, enabling passive control of COM trajectory. The simulation shows that such control, which utilizes gravity to improve efficiency, is achieved through optimization. These results suggest that the observed predictive behavior can be explained within the framework of model predictive control.

Several studies have suggested that model predictive control (MPC), also known as receding horizon control, can reproduce human behavior. For example, intermittent behavior observed during slow finger movements has been explained as repeated short-term control within an MPC framework[32,33]. In a cursor tracking task with a limited display range, participants' movements were also modeled using MPC, and it was noted that the prediction and optimization horizons increased with task proficiency[34]. In the field of postural control, quiet standing has been successfully modeled using MPC, with arguments emphasizing its energy efficiency benefits[44]. We previously reproduced predictive movements in bipedally standing rats using MPC and estimated the prediction period based on their movement patterns[45]. However, the aforementioned studies, including our work on

standing rats, did not address active preparatory movements, such as shifting the COM in advance of external disturbances. The anticipatory movements based on prediction, as observed in the present experiments, have not been explained in conventional postural control research, and addressing them constitutes a key contribution of the present study.

We next consider whether MPC is an appropriate framework compared to conventional posture control models. In studies of standing posture control, two types of control mechanisms have been proposed: stiffness control based on intrinsic muscle stiffness[46], and sensory feedback control[24,47]. Furthermore, based on the noise and frequency characteristics observed in human quiet standing, intermittent control has been proposed as a mechanism for sensory feedback regulation[35–37]. In quiet standing, intermittent control is thought to involve switching control on or off depending on the estimated state's frequency content[36] or whether it exceeds a sensory threshold[40], potentially involving cortical contributions[48]. The intermittent nature of the control input corresponds to control being executed in short bursts or intervals[32], and such switching requires internal state

prediction. These features align well with the principles of model predictive control[30].

Compared to conventional models of quiet standing based on feedback control, a key feature of MPC is its use of sequential optimization using an internal model. Control models incorporating internal models to generate posture control inputs with delay compensation have been proposed since the 1990s[49]. However, subsequent studies demonstrated that simpler feedback control models could adequately explain the stability and characteristics of human COM sway[50,51]. Furthermore, instances in which muscle activity precedes movement have been reported[52], raising expectations for predictive control mechanisms. However, such anticipatory activity has also been reproduced by incorporating high-gain velocity feedback[53,54], indicating that feedback control can account for most phenomena in quiet standing without requiring predictive control based on internal models. As a result, internal models have received less emphasis in explanations of control input generation during quiet standing. On the other hand, there is also evidence that internal models and their impairments affect quiet standing behavior[55]. In this context, state estimation is a function that relies on internal models to support posture control. In fact, estimating the state necessary for postural regulation involves integrating sensory information from vestibular, visual, and somatosensory systems. Optimal state estimation using Kalman filters, which are based on internal models, has been shown to effectively represent this integration[25–27]. In other words, internal models are crucial when considering state estimation in posture control. Since accurate estimation or prediction of the state is essential to account for the predictive movements examined in this study, the inclusion of internal models in postural control becomes indispensable. From this perspective, model predictive control, which generates control inputs based on internal models, can be regarded as a natural extension of previous models of posture control.

One of the key contributions of this study, which models predictive posture control using MPC, is that it explains control through the GC and demonstrates that the strategy of utilizing gravity for posture adjustment emerges as an outcome of optimization. The fact that this strategy was observed in both experiments and simulations suggests that humans employ a skillful mechanism for controlling COM by leveraging gravity with minimal energy expenditure. Such passive movements that exploit gravity have not been emphasized in previous posture control studies. However, the role of gravity in movement generation has been considered important in other types of human motion, particularly in gait control. During walking, leg movements are organized to alternate between kinetic energy from leg swing and potential energy due to gravity. Energy lost through collisions is compensated for by muscle activity[56]. This cyclical energy exchange results in a limit cycle that contributes to locomotor stability. In fact, studies have shown that walking based on passive dynamics, without the use of active force, can produce stable motion[57,58]. It has also been noted that such passive locomotion resembles human walking in both kinematic features[59] and energy characteristics[60,61]. These findings indicate that humans are capable of harnessing gravity when generating movement, and the present study demonstrates that this mechanism also plays a role in predictive posture control during quiet standing.

While the model predictive control-based model used in this study successfully reproduced the COM shift and GC activity observed prior to floor tilt, it failed to account for the impulse-like GC activation during the tilt itself based on MPC alone. To further examine the impulse-like GC activity observed after tilt onset, we analyzed its temporal characteristics and found that it differed from the control properties observed during the predictive period. Specifically, whereas the forward COM shift and GC activity during the predictive period varied depending on the presence or absence of the cue, no significant difference was observed in the latency of GC activation after tilt onset between the cue and no cue conditions. This result suggests that muscle activity following tilt onset may be generated by mechanisms distinct from the cue-dependent predictive control operating during the predictive period. Indeed, in upper-limb movements, long-latency responses (LLRs) have been modeled as feedback processes incorporating

prediction[62], whereas in upright postural control, LLRs have been reported to behave as preprogrammed signals triggered by muscle stretch[63], with the early component in particular exhibiting preprogrammed characteristics[64]. Based on these findings, we extended our model by incorporating an additional term representing reflexive activity following floor tilt, in addition to the MPC responsible for predictive control. With this extension, the impulse-like GC activation during the floor tilt and the emergence of TA activity were reproduced, while the overall postural behavior remained stable under the combined action of predictive control and reflexive input (Fig. 9). Although further discussion is required regarding how LLRs during floor tilt should be modeled, the present results demonstrate that the proposed postural control model based on MPC can be extended to describe not only anticipatory adjustments but also responses during the perturbation itself.

In the present study, based on the experimental design of Kolb et al.[9,65], only backward tilt perturbations were applied to investigate predictive muscle activity prior to floor tilt. In studies of postural control, the activity of the plantarflexor muscle group, particularly the GC, has been consistently used as a primary indicator of postural reflexes and predictive activity in response to floor tilt perturbations[12,66]. On the other hand, it cannot be excluded that restricting perturbations to a single direction influenced the participants' postural configuration. In particular, if participants came to expect only backward tilts, they might have adopted a slightly forward-leaning posture compared with the initial phase of the experiment, when the perturbation direction was not yet anticipated. However, as shown in Fig. 3, no clear systematic differences were observed between the early trials (trials 1–5) and the later trials. This result suggests that the influence of unidirectional perturbations on posture and predictive muscle activity was limited within the range observed in the present study.

Our experiments revealed anticipatory activation of the GC prior to floor tilt, whereas Kolb's studies[9,65], reported anticipatory activity in the tibialis anterior (TA). This discrepancy may be attributed to differences in the prediction period. Kolb's study used a prediction interval of approximately 450 ms, whereas the present study used a period of about 2 s. Our findings indicate that GC activity supports the body against gravitational forces. Utilizing gravity to initiate body movement requires sufficient time, while faster movements necessitate active muscle activation. Indeed, this study shows that TA activity emerges when the external input is small (Fig. 7) or when the COM is located posteriorly, making it more difficult to exploit gravity (Fig. 8B).

Elucidating the mechanisms underlying predictive postural control is of significant importance for understanding the factors contributing to postural dysfunction in neurological disorders and for developing rehabilitation strategies. In environments with cues and floor tilting similar to those used in the present study, patients with diffuse cerebellar pathology have been shown to lack predictive postural responses[9]. Similarly, rats with lesions in the cerebellar vermis exhibited a significant decline in their ability to maintain posture in response to tilt disturbances following a cue during bipedal standing[67]. Existing posture control models have not accounted for such impairments in cerebellar function. The cerebellum receives error signals during movement via the inferior olive nucleus[68,69] and uses this information to construct internal models[70,71]. Dysfunction in the formation or utilization of these internal models is considered a key contributor to the motor impairments observed in cerebellar disorders. Therefore, to simulate dysfunction associated with cerebellar impairment, it is essential to employ posture control models that generate control inputs based on internal models. This has been difficult to address using conventional posture control models, but can be more naturally examined within the MPC framework used in the present study. In summary, the posture control framework based on MPC proposed in this study offers a promising approach to explaining postural dysfunctions associated with cerebellar disorders. Moreover, it is expected to provide valuable insights for the development of effective treatment and rehabilitation strategies.

## Methods

### Experimental design

This study aimed to investigate whether humans generate predictive postural adjustments prior to expected perturbations, and to examine the underlying control mechanisms. We combined human experiments and computational simulations to evaluate both the existence and origin of anticipatory body movements in response to predictable external disturbances.

In the experiment, subjects stood on a motor-controlled floor platform that introduced backward tilting disturbances. Two conditions were tested: one with an auditory cue given 2–3 s before the tilt, and one without. This within-subject design enabled direct comparison of anticipatory responses under cued and uncued conditions, with a focus on forward shifts of the COM and associated muscle activity. To interpret the control strategies underlying the observed behavior, we developed a computational musculoskeletal model equipped with model predictive control (MPC), which optimizes future control inputs based on internal predictions of body dynamics. By simulating the same perturbation conditions as in the human experiment, we tested whether anticipatory COM shifts and selective muscle activation could emerge from optimizing predicted movement.

This integrative approach allowed us to both characterize anticipatory postural behavior in humans and evaluate whether a predictive optimization strategy could account for such behavior, thereby bridging empirical observations with theoretical modeling.

### Experimental procedure

The subjects were given a cue to tilt the floor, and their predictive movements before the tilt were measured. The experimental subjects were ten healthy individuals (five males and five females) in their 20–50 s, with heights (mean ± standard deviation (SD)) of 164.6 ± 8.1 cm and body masses (mean ± SD) of 62.6 ± 9.1 kg (Supplementary Table S6). In the experiment, the subjects were asked to stand on the experimental apparatus (floor) for 30 s, and their responses to the floor tilt disturbance were measured. While standing, the subjects were instructed to cross their arms in front of their chest to ensure that the motion capture markers were not obscured. The experiment began with the floor in a horizontal position. Twenty seconds after the start of the experiment, the floor began to tilt at a rate of approx. 4 degrees per second and stopped when the tilt angle reached 4 degrees. The floor was operated by a servo motor, and the tilt angular velocity of the floor was kept constant during the operation. The actual floor tilt duration measured by the motion capture system was 0.81 (0.03) s on average (SD) for all subjects. The experiment was conducted under two conditions: with a cue provided by a buzzer sound given 2–3 s before the start of the tilt, and without a cue. The start time of the cue was set by the operator manually pressing a button after seeing the time 3 s before the start of the tilt, introducing some variation and not setting the cue time completely fixed. The cue sound stopped after the tilt ended. After 30 trials without cues, 30 trials with cues were conducted for each subject. During the experiment, subjects closed their eyes and wore ear muffs. The buzzer sound was loud enough to be heard even with the ear muffs on. The above experimental conditions were based on Kolb's experiment[9,65], which examined responses to tilt disturbances while standing. However, to focus on responses during the cue period, the cue period and stimulus period were lengthened.

The subjects' movements were measured using a motion capture system (Qualisys Miqus M3, eight cameras). The measurement frequency was 300 Hz. For the measurements, 11 motion capture markers were attached to the subjects' heads, both shoulders, greater trochanter, knees, ankles, and toes, and markers were also attached to the four corners of the floor of the experimental apparatus. Using the motion data of the markers attached to the subjects, the COM movements were calculated. From the marker data, the COM movements of the upper body, thigh, lower leg, and foot segments were calculated, and based on the weight ratios of each segment[72], the COM movements of the entire body were calculated. The COM position is defined as the horizontal position of the COM in the global coordinate system with the ankle as the origin. The floor tilt angle, ankle angle, and hip angle are defined as the relative angles to the horizontal plane, the angle perpendicular to the floor, and the angle from the lower leg angle, respectively (Fig. 2H). To construct trajectories of COM position and velocity during movement (COM phase portraits[38,39]), the COM velocity time series was derived by numerically differentiating the COM position time series. Prior to numerical differentiation, the COM position data were low-pass filtered at 5 Hz using a fourth-order Butterworth filter. Numerical differentiation was then performed using a five-point differentiation method.

Muscle activity was measured using a wireless electromyograph (Cometa PicoEMG). The measurement frequency was 2000 Hz and was input to an analog board synchronized with the motion capture system and recorded at 300 Hz. The measurement points were 10 locations: the left and right tibialis anterior, gastrocnemius, rectus abdominis, erector spinae, right iliopsoas, and right gluteus maximus. The muscle activity data were filtered using a 1 Hz high-pass filter and a 10 Hz low-pass filter (both fourth-order Butterworth filters), and the muscle activity percentage (percentage MVC) was calculated using the maximum muscle activity measured after completion of the floor tilt experiment. Note that, among the ten muscles measured, activity was barely detected except for the muscles around the ankle (TA and GC), so the analysis focused on the TA and GC. Additionally, the average values of the TA and GC on both sides were used for the analysis.

### Musculoskeletal modeling and simulation

To investigate what control law constitutes predictive motion, we constructed a mathematical model consisting of a body musculoskeletal model and a control model with model predictive control (Fig. 5). The body model consists of two links, with two joints at the ankle and hip, and is modeled so that changes in the angle of the floor act as external forces (see "body model" in Supplementary Methods). The values of each body parameter were taken from a quiet standing study using a two-link body model[73] (Supplementary Table S7A). Biological noise was modeled as Gaussian white noise with a magnitude of 0.33 Nm[74], generated as separate series for the hip and ankle, respectively. The initial angles of the ankle and hip were set to −0.11 and 0.18 rad, respectively, based on the mean values observed at the cue onset (CS) in trials with a cue: ankle −0.11 ± 0.02 rad; hip 0.18 ± 0.09 rad.

As a muscle model, four muscles were arranged: the tibialis anterior (TA) and gastrocnemius (GC) muscles around the ankle, and the iliopsoas (IL) and gluteus maximus (GM) muscles around the hip. The dynamical model of the muscle consists of contraction elements that respond to control inputs and passive elastic elements, and generates muscle force according to the force-length and force-velocity relationships in these elements[75,76] (see "muscle model" in Supplementary Methods). The values of each muscle parameter are shown in Supplementary Table S7B. The simulation duration was set to 20 s, and a tilt disturbance of 4 degrees was applied to the floor for 1 s between 15 and 16 s. Simulations were computed using the Euler–Maruyama method for solving stochastic differential equations to account for noise effects. The simulation frequency was 1000 Hz.

### Control model

To maintain stable posture and generate predictive movements, a control system was constructed using model predictive control (MPC). MPC is a control method that repeatedly predicts and optimizes near-future states. When generating control inputs, MPC predicts the states up to $N_P$ steps after the current time by changing the control input for the interval of $N_M$ steps, and calculates the control input such that the predicted states during the prediction period are optimal. Of the calculated control inputs, only the first step is used as the actual control input. Here, $N_P$ is referred to as the prediction horizon, and $N_M$ is referred to as the control horizon. In this study, the frequency of prediction optimization by MPC was set to 100 Hz, and $N_P$ was set to 30 steps (3 s) to include the interval from the start of the cue to the end of the tilt. $N_M$ was set to 2 steps because, in a previous posture control study using MPC[45], the results were nearly the same when $N_M$ was set to 2 steps compared to more than 2 steps. The internal model used for

prediction in this study is assumed to be a model that has been sufficiently learned and can adequately reproduce the body and environment, and thus is considered to correspond to the equations of motion of the body. In other words, the equations of motion of the body and muscles were directly used as the internal model. Note that the noise in the equations of motion of the internal model was set to 0.

The control inputs generated by MPC are muscle activity levels at the ankle and hip (see "muscle model" in Supplementary Methods). For inputs around the ankle, muscle activity levels for TA and GC are calculated separately. For control inputs around the hip, to reduce computational cost, if the muscle activity level is positive, the input is applied only to IL, and if it is negative, the input is applied only to GM (Fig. 5). The evaluation function for optimizing the prediction period is set to minimize the horizontal displacement of the COM $x_{COM}$-$x_{ref}$ control input $u$, and the change in control input $\delta u$, that is, $J = \Sigma_{Np}\{(x_{COM}-x_{ref})^2 + (w_u\ u)^2 + (w_{\delta u}\ \delta u)^2\}$. The target value for the COM position $x_{ref}$ is 61 mm, which drives from the average horizontal position of the COM for all subjects at CS in the experiment with the cue. When evaluating the effect of the COM reference position (Fig. 8A–C), this value was varied. $w_u$ is the weight of the input, which was set to 0.01, and this value was varied when evaluating the input weight (Fig. 7). $w_{\delta u}$ is the weight of the change in control inputs, which was set to 1. The weights for the input and input changes of the three muscles (TA, GC, and IL or GM) around the ankle and hip were all set to the same value. Sensory input reaches the MPC with a delay, which was set to 150 ms[74]. The MPC was implemented using the nlmpc function in the Model Predictive Control Toolbox of Matlab.

Another control element contributing to stability during standing is the passive effect of body stiffness, which plays a stabilizing role in cooperation with the stretch reflex[46]. This mechanism generates the torques to the ankle and hip in the manner of proportional and differential control (see "stiffness control" in Supplementary Methods). The values of these control parameters are summarized in Supplementary Table S7C.

To reproduce muscle activity during floor tilt, additional simulations were performed using an extended model in which a term representing reflexive activity was added to the control input based on MPC. Specifically, a constant activation (50%) was applied to the GC during a 100-ms window starting 100 ms after floor tilt onset. To display the GC results, the simulated GC activity was filtered using the same 10-Hz low-pass filter (fourth-order Butterworth filter) as applied to the experimental data.

### Ethical considerations

The experimental protocol was approved by the Ethics Committee of the University of Electro-Communications (approval no. 22029). All participants received the written experimental protocol and signed an informed consent form prior to study initiation in accordance with the Declaration of Helsinki. All ethical regulations relevant to human research participants were followed.

### Statistics and reproducibility

The human experiment included ten biologically independent participants. Each participant performed 30 trials in the no-cue condition and 30 trials in the cue condition, with rest between trials. For analyses other than the trial-stage analysis, the first 10 trials in each condition were excluded, and the remaining 20 trials were analyzed. For trial-stage analyses, all 30 trials in each condition were used and divided into six stages of five trials each. In the simulation, repeated runs with different random noise realizations were treated as independent simulation trials; unless otherwise stated, 20 simulation trials were performed for each condition.

Changes in movement and muscle activity with and without auditory cues were evaluated for each subject using a t-test. To evaluate changes in muscle activity during the cue period (between CS and FS), linear regression was applied to each subject's muscle activity during this period, and the resulting regression coefficients were calculated. The MATLAB fitlm function was used to compute the linear regression. To examine whether the extent of the COM shift and changes in muscle activity during the cue period

varied depending on cue presence and trial stage, a two-way ANOVA was performed to test the effects of these two factors. In addition, a one-way ANOVA was conducted on the trial stage within each cue condition. For the simulation results, movements and muscle activity at CS and FS were compared using t-tests to investigate whether the activity reflected predictive control. Furthermore, to determine whether the amount of preceding COM shift varied depending on the COM reference position, a one-way ANOVA was performed. To examine whether the impulse-like GC activity observed after floor tilt differed between the cue and no-cue conditions, the time from floor tilt onset to the first peak of GC activity was calculated. This peak latency was analyzed using a two-way ANOVA with subject and cue condition as factors. All statistical tests were two-sided. Exact $p$-values are reported where possible.

### Reporting summary

Further information on research design is available in the Nature Portfolio Reporting Summary linked to this article.

### Data availability

The measured data from human experiments, including joint angles recorded by the motion capture system, COM positions calculated from the motion capture data, EMG data of the TA and GC muscles, are available on GitHub and archived on Zenodo[77]. All other data are available from the corresponding author on reasonable request.

### Code availability

The program code used for the analysis and simulation is available on GitHub and archived on Zenodo[77]. The archived version corresponds to the version used in this study. No restrictions apply to access. The software was developed and tested using MATLAB R2024b and requires the Model Predictive Control Toolbox and the Statistics and Machine Learning Toolbox.

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

## Acknowledgements

This work was supported by Grants-in-Aid for Scientific Research (B) (no. 24K00833) and Scientific Research on Innovative Areas (no. 19H05728), funded by the Ministry of Education, Culture, Sports, Science, and Technology of Japan.

## Author contributions

T.F., D.Y., and A.K. designed the concept and planned the experiment. T.F., M.O. performed the experiment. T.F., M.O. analyzed the data and performed the computer simulation. T.F., D.Y., and A.K. prepared the manuscript. All authors reviewed the manuscript.

## Competing interests

The authors declare no competing interests.

## Additional information

**Supplementary information** The online version contains Supplementary material available at https://doi.org/10.1038/s42003-026-10016-2.

