## [Transparent Peer Review File · Communications Biology]

Anticipatory Postural Control Emerges from a Predictive and Optimized Strategy for Movement Preparation

Corresponding Author: Dr Tetsuro Funato

Version 0:

Reviewer comments:

Reviewer #1

(Remarks to the Author)

The study investigates how humans prepare their posture before predictable disturbances, such as backward floor tilts. Normally, people shift their center of mass (COM) forward before the tilt, accompanied by activation of the gastrocnemius (GC) muscle, even though this muscle typically generates backward torque. This anticipatory adjustment cannot be explained by simple reflex feedback. To explain this, the researchers developed a computational model using Model Predictive Control (MPC), which successfully reproduced the observed behavior. I fully support the publication of this article.

Reviewer #2

(Remarks to the Author)

Brief summary:

Previous work has shown that when postural perturbations are predictable, individuals make adjustments prior to the upcoming disturbance to minimize its destabilizing effects. In the case of a perturbation known to induce backward postural sway, such as a forward surface translation or backward (toes up) platform rotation, individual adopt an anticipatory forward lean. Funato and colleagues attempted to determine if this anticipatory forward lean was part of a predictive and optimized control strategy. Experimental data demonstrated that an anticipatory forward lean was adopted prior to conditioned (or cued) backward platform rotations, and this was accompanied by increased activity of the medial gastrocnemius. Simulations using a predictive model controller replicated some features of this behavior, although not as well as the authors make it seem (as discussed below), leading them to conclude that anticipatory postural adjustments arise from optimizing predicted future states.

While I appreciate work combining both experimental and simulation data, there are several critical issues with this study that I believe make it unacceptable for publication in its current form.

Major issues (no particular order):

1. As far as I can tell, the experimental data reveal nothing new and an otherwise well-documented phenomenon (i.e., increased medial gastrocnemius activity accompanying a forward lean) is characterized as something 'contradictory' to justify the need for neuromechanical modeling.

Specifically, the authors state: "It is noticeable that an increase in GC activity occurs during forward movement of the COM. GC is located posterior to the ankle, its activation induces a backward ankle torque, thereby shifting the COM backward. However, since the COM actually moves forward, this appears to be contradictory behavior." This behavior is not contradictory and should not be characterized as such. When standing normally, the vertical projection of the COM falls a little in front of the ankle, meaning that there is a gravitational torque acting to pull the COM forward. Since passive stiffness at the ankle isn't sufficient to counteract this gravitational torque, active torques generated from plantar flexors are needed to prevent the COM from toppling forward. Naturally, if the COM is maintained in a more forward position, greater counteractive torques will be needed to stay upright. Thus, the increased activation of the GC is expected.

2. Several analyses applied to the experimental data are misleading or insufficiently interpreted.

First, the authors calculate a co-contraction index (CCI) using the methods of Falconer and Winter. From their data, they find that people who tend to adopt a larger anticipatory lean have lower a CCI compared to those who do not lean forward. They suggest that this reflects two distinct activation strategies, with those latter subjects adopting a co-contraction strategy rather than a lean. However, because the CCI calculation is a ratio of normalized EMG and the TA is generally quiescent, the CCI value will vary inversely with GC activity which is closely linked with the magnitude of the forward lean.

Second, the linear regression model applied to the relationship between COM position and MG activity characterize how these variables are related only on longer time-scales, giving the impression of a 'contradictory' relationship as the authors describe. Activation of the MG is associated with forward COP movement, which causes backward acceleration of the COM. Thus, examining the relationship between MG activity and horizontal COM acceleration would reveal a less contradictory relationship and help to more fully explain how these variables interact.

Third, and related to the above point, more sophisticated analyses of the relationship between the MG and COM movement could have illustrated how the COM started to move forward in the experimental data. Instead of simply looking at mean GC and TA activity in the time between CS and FS, the authors could have attempted to generate waveform averaged EMG aligned to the onset of forward COM movement. Presumably, this would have revealed an initial suppression of GC activity that permitted gravitational-induced forward COM shift.

2. While I laud the authors for combining their experimental data with a neuromechanical model, there are several results related to their model that make we question its validity.

(A) In the analyses where the authors adjust the weight of control inputs to examine how changes in the efficiency of the predictive controller changes the anticipatory lean, it doesn't look as though the magnitude or profile of the lean between CS and FS changes at all. Rather, it looks to stay the same, but the baseline target position changes, resulting in a larger absolute lean. This same observation applies to the GC activity. This invariant behavior is not discussed by the authors, but is instead characterized on the basis of the absolute forward position and GC activity. (B) Furthermore, in subsequent analyses where they manipulate the initial target position, such that the COM some times starts well behind the ankle joint (-50 mm), the magnitude of the preparatory lean between CS and FS doesn't change. This is somewhat baffling and is unlikely to be replicated by experimental data since a 20-30 mm lean starting a from a -50 mm position still leaves the individuals' COM behind the ankle joint and, therefore, in a very precarious position if they're expecting a backward platform tilt. (C) While the authors offer a possible explanation for the complete absence of TA activity and conspicuous increase in GC activity following platform tilts in their simulations, it does not inspire confidence in the other aspects of their model. This is all to say that the validity of the neuromechanical model is dubious and the authors claim that the results from their neuromechanical match their experimental data is somewhat misleading.

Minor issues (no order):

- Were only backward tilt perturbations applied? If so, why was this done? Presumably participants would have adopted a slight pitched forward position knowing that the perturbations could only come in one direction, which may have blunted additional anticipatory leaning following the CS (as was seen in some subjects).
- In the introduction when discussing automatic postural responses (lines 44-48), the direction of the translational disturbance should be made clear.

Version 1:

Reviewer comments:

Reviewer #2

(Remarks to the Author)

I'd like to thank the authors for carefully considering my criticisms of their previous draft. They've made considerable changes to the paper, including the addition of several new figures/analyses that I think have greatly improved their manuscript. I have only two minor suggestions for them to consider:

The authors use a bit of a strawman argument to justify their study. At points in the abstract, introduction, and discussion, it is described as though they want to know if the anticipatory (predictive) postural adjustment prior to predictable perturbations are achieved through predictive rather than feedback control mechanisms. This is self-evident as these adjustments are predictive in their nature and therefore cannot be produced through feedback control. I believe this rationale should be modified to emphasize the optimal control aspect of their work rather than contrasting predictive and feedback control.

Some discussion of what the center of pressure (COP) is doing during the anticipatory postural adjustments would be nice. The authors discuss the COM-COM velocity phase plane analyses (which I appreciated that they added to the paper) and note how the GC activity is generally greatest when the COM is in a forward position while moving forward. They then suggest that mechanisms other than muscle-generated torques are contributing to the preparatory forward COM movement (lines 199-216). Here, the authors make a statement which makes it seem as though this behavior is somewhat unexpected: "However, when focusing on the fact that GC can generate only posterior ankle torque while the COM is being shifted forward, the direction of the generated torque is opposite to the direction of the intended COM shift. This inversion between the direction of control and the direction of muscle-generated torque suggests that forces other than muscle-generated torque contribute to moving the COM forward."

What's happening during these points in time is the increased GC activity is causing the COP to move forward ahead of the COM to slow its acceleration. This is why the GC is highly active while the COM is still moving forward. This is all part of what has been referred to previously as a 'drop and catch' pattern that occurs repeatedly during quiet standing during forward sways (see work by Loram and colleagues between 2001-2006). I.e., first, the GC relaxes, this causes the COP to fall behind the COG and allows the COM to accelerate forward (the drop). As the COM moves forward, the GC then turns on to slow the COM's acceleration and stop it in its new position (the catch). Adding a small statement of these COP-COM dynamics would be helpful to clarify what's happening during their forward sways.

Response to Reviewer #1

The study investigates how humans prepare their posture before predictable disturbances, such as backward floor tilts. Normally, people shift their center of mass (COM) forward before the tilt, accompanied by activation of the gastrocnemius (GC) muscle, even though this muscle typically generates backward torque. This anticipatory adjustment cannot be explained by simple reflex feedback. To explain this, the researchers developed a computational model using Model Predictive Control (MPC), which successfully reproduced the observed behavior. I fully support the publication of this article.

We thank the reviewer for the positive and supportive evaluation of our work.

We are pleased that the reviewer found our interpretation of the anticipatory postural control and the use of the model predictive control framework to be appropriate.

Response to Reviewer #2

Brief summary:

Previous work has shown that when postural perturbations are predictable, individuals make adjustments prior to the upcoming disturbance to minimize its destabilizing effects. In the case of a perturbation known to induce backward postural sway, such as a forward surface translation or backward (toes up) platform rotation, individuals adopt an anticipatory forward lean. Funato and colleagues attempted to determine if this anticipatory forward lean was part of a predictive and optimized control strategy. Experimental data demonstrated that an anticipatory forward lean was adopted prior to conditioned (or cued) backward platform rotations, and this was accompanied by increased activity of the medial gastrocnemius. Simulations using a predictive model controller replicated some features of this behavior, although not as well as the authors make it seem (as discussed below), leading them to conclude that anticipatory postural adjustments arise from optimizing predicted future states.

While I appreciate work combining both experimental and simulation data, there are several critical issues with this study that I believe make it unacceptable for publication in its current form.

Thank you very much for your careful and constructive review.

In response to the reviewer's comments, we have substantially revised the manuscript by reorganizing the relationship between postural state (COM dynamics) and muscle activity during the predictive period, based on both experimental and simulation results. These revisions were made to clarify aspects of predictive postural control that are difficult to explain within the framework of conventional postural control theories. The main revisions are summarized below.

- To clarify how the relationship between the direction of COM displacement and GC activity during the predictive period arises, and to identify the limitations of existing control frameworks, we added a new subsection in the Results section entitled **“What Control Mechanism Can Explain the Forward Shift of the COM Accompanied by Posterior GC Activity?” (Line: 188-189)**. In this subsection, we explicitly state that this issue cannot be fully explained by conventional postural control theories.
- In accordance with the reviewer's suggestion, we **removed the results and discussion related to the co-contraction index (CCI)** and replaced them with a new figure (**Fig. 2G**) illustrating the relationship between COM dynamics and GC activity. In addition, we added the corresponding simulation results as **Fig. 6G**.
- We added **Fig. 4** to illustrate the relationship between the COM phase portrait (COM position-

velocity relationship) and GC activity during the predictive period, thereby clarifying the postural states in which GC activation occurs. The corresponding simulation results are presented in Fig. 6H.

- For the simulation results (Figs. 6 and 7), we added analyses of CS-FS differences and regression coefficients for COM motion, joint angles, and muscle activity, analogous to the analyses performed for the experimental data, allowing for direct comparison between experimental and simulation results.
- We clarified that the initial posture and the target posture were set to be identical in the simulations and explicitly stated the rationale for this choice.
- We examined the conditions required to reproduce the behavior during floor tilt in the simulations and demonstrated that the GC-activity after tilt onset followed by the emergence of TA activity can be reproduced by extending the mathematical model by adding tilt-triggered activity corresponding to long latency response (New subsection “Extension of the model to reproduce reflexive activity during floor tilt” in the Result section and new Figure 9).
- We added a discussion of the potential effects of restricting the perturbation direction to a single direction in the experimental design and revised the Introduction to clarify the direction of the applied perturbations.

Detailed responses to each specific comment are provided below.

Major issues (no particular order):

1. As far as I can tell, the experimental data reveal nothing new and an otherwise well-documented phenomenon (i.e., increased medial gastrocnemius activity accompanying a forward lean) is characterized as something ‘contradictory’ to justify the need for neuromechanical modeling. Specifically, the authors state: “It is noticeable that an increase in GC activity occurs during forward movement of the COM. GC is located posterior to the ankle, its activation induces a backward ankle torque, thereby shifting the COM backward. However, since the COM actually moves forward, this appears to be contradictory behavior.” This behavior is not contradictory and should not be characterized as such. When standing normally, the vertical projection of the COM falls a little in front of the ankle, meaning that there is a gravitational torque acting to pull the COM forward. Since passive stiffness at the ankle isn’t sufficient to counteract this gravitational torque, active torques generated from plantar flexors are needed to prevent the COM from toppling forward. Naturally, if the COM is maintained in a more forward position, greater counteractive torques will be needed to stay upright. Thus, the increased activation of the GC is expected.

Thank you very much for your insightful comments. As the reviewer correctly pointed out, the phenomena observed between COM motion and GC activity in the experiments presented in this manuscript do not involve any physical inconsistency. We fully agree with the reviewer that, from a purely physical perspective, no contradiction exists. We recognize, however, that our original manuscript did not clearly state **with respect to what assumption** we perceived a “contradiction.” In this regard, the reviewer’s comments prompted us to carefully reconsider the relationship between COM motion and GC activity, and to clarify what kind of control framework leads to feel the inconsistency. We sincerely appreciate the opportunity to revisit and refine this point.

We have now clarified that the perceived “contradiction” does not arise from the phenomenon itself, but rather from the assumption that postural control is achieved solely through conventional feedback control, as has been traditionally considered. Under a feedback control framework, if the COM is to be shifted forward during the predictive period, the target position must be set anterior to the current COM position. Such feedback control would generate a forward-directed torque as a control command. If this forward torque command were directly translated into muscle activation, it would primarily result in activation of the tibialis anterior (TA). In other words, this framework would predict TA activation rather than the GC activity observed experimentally.

What is missing in this line of reasoning, as highlighted by the reviewer, is the contribution of gravitational torque. If gravitational torque is incorporated into the generation of control torque, such

that a portion of the required forward torque is provided by gravity and the remainder by muscle commands, it becomes possible to shift the COM forward while maintaining stability through the combined action of gravitational torque and GC activity. To our knowledge, however, conventional postural control models have not explicitly exploited gravitational torque as an active component of movement generation in this manner.

One reason for this may be that the magnitude of gravitational torque varies continuously with posture, making it difficult to estimate. Estimating gravitational torque therefore requires an internal model that represents the relationship between posture and torque. In the model predictive control framework proposed in this study, state variables are estimated using an internal model, and optimal control inputs are generated accordingly. From this perspective, the controller can be interpreted as estimating gravitational torque and generating GC activity to compensate for the remaining torque required to maintain stability. We believe that this mechanism explains a phenomenon that could not be accounted for by conventional feedback control without an internal model, and that it is reproduced here for the first time using the proposed approach.

We acknowledge that this line of reasoning was not sufficiently articulated in the original manuscript. In response, we have added a new subsection within the Results section, positioned between the experimental and simulation results, entitled “**What Control Mechanism Can Explain the Forward Shift of the COM Accompanied by Posterior GC Activity?**” (Line: 188-189). In this subsection, we (1) summarize the relationship between COM motion and GC activity pointed out by the reviewer, and (2) explain why a control framework incorporating an internal model is necessary to account for this phenomenon. Specifically, we have added the following description to the manuscript.

Line 190-198:

When comparing the direction of the COM (Fig. 2A) and muscle activity (Fig. 2E), it is noticeable that an increase in GC activity occurs during forward movement of the COM. To examine how GC activity relates to COM shift, we plotted the relationship between the magnitude of COM shift during the cue period and the rate of change in GC activity over the same period (Fig. 2G). The results showed that GC activity increased as COM shift increased (linear regression: $R^2 = 0.75$, $P=0.001$). The GC muscle is located posterior to the ankle joint, and its activation generates a posterior ankle torque. This figure suggests that as the COM moves further forward, GC activity increases to counteract this displacement and maintain postural balance.

Line 210-229:

The relationship between COM motion and GC activity described above is not inconsistent with the role of GC activity in maintaining postural stability. However, when focusing on the fact that GC can generate only posterior ankle torque while the COM is being shifted forward, the direction of

the generated torque is opposite to the direction of the intended COM shift. This inversion between the direction of control and the direction of muscle-generated torque suggests that forces other than muscle-generated torque contribute to moving the COM forward.

During upright standing, gravity is continuously acting on the body, and as posture becomes more forward-leaning, gravity generates a stronger forward-directed torque. By simultaneously utilizing the forward torque generated by gravity and the posterior torque generated by GC activity, it becomes possible to shift the COM forward while maintaining stability. However, implementing this strategy requires estimating the gravitational torque and generating GC activity based on the difference between the gravitational torque and the desired net torque. Because the magnitude of gravitational torque varies continuously with posture, estimating this torque necessitates an internal model that represents the relationship between posture and torque. Conventional feedback control used in postural control do not incorporate such internal-model-based state estimation within the control system, making it difficult to explain the present phenomenon. In contrast, control strategies that generate control inputs based on an internal model of the body and explicitly account for gravitational torque may be able to reproduce this behavior.

Among these additional descriptions, point (1) in particular was developed based on the reviewer's subsequent comment, in which we explicitly evaluated and described the relationship between COM motion and GC activity (Fig. 2G). As a result, as the reviewer correctly pointed out, we found a clear relationship in which GC activity increases with forward COM inclination (linear regression: $R^2 = 0.75$, $P = 0.001$). This relationship was also evaluated in the simulation (Fig. 6G), where a similar pattern was observed. We will further elaborate on this point in a later response.

In addition, as the reviewer noted, the term "contradiction" used in the Abstract could lead to misunderstanding. We have therefore revised the Abstract to remove the word "contradiction."

We believe that these revisions clarify both the phenomena observed in the experiments and the specific contributions that the proposed model aims to elucidate. Thank you very much for your valuable comments.

2. Several analyses applied to the experimental data are misleading or insufficiently interpreted. First, the authors calculate a co-contraction index (CCI) using the methods of Falconer and Winter. From their data, they find that people who tend to adopt a larger anticipatory lean have lower a CCI compared to those who do not lean forward. They suggest that this reflects two distinct activation strategies, with those latter subjects adopting a co-contraction strategy rather than a lean. However, because the CCI calculation is a ratio of normalized EMG and the TA is generally quiescent, the CCI value will vary inversely with GC activity which is closely linked with the magnitude of the forward lean.

Thank you very much for your valuable and insightful comments. We found your observation that the extremely low level of TA activity leads to an inverse correlation between the co-contraction index (CCI) and GC activity, and that this causes the CCI to no longer purely reflect co-contraction, to be particularly perceptive.

We agree with the reviewer's assessment regarding the calculation of the CCI. Under the present experimental conditions, TA activity was consistently low, and therefore we concluded that using the CCI to evaluate co-contraction was not appropriate. In addition, with respect to our original claim that a co-contraction strategy might be employed, we acknowledge that, as the reviewer correctly pointed out, the CCI does not adequately capture co-contraction in this case. Accordingly, we agree that such a claim should not have been made.

In this manuscript, the characteristics of this group of participants (Cluster 2) are not a central focus. Rather, the main objective of the study is to explain the control mechanism underlying the predictive forward COM shift observed in the participant group exhibiting greater COM advance (Cluster 1). For this reason, we determined that the discussion regarding co-contraction should be removed, and we have deleted this description from the manuscript.

Specifically, we removed the results and methodological descriptions related to the CCI from both the Results and Methods sections. In addition, as described in our previous response, we added a new figure illustrating the relationship between COM motion and GC activity (new Fig. 2G) and included a corresponding analysis and discussion.

We believe that these revisions allow the manuscript to focus more clearly on its primary objective, which is to elucidate the mechanism of COM shift during the predictive period, and thereby improve the overall clarity of the argument. Once again, thank you very much for your valuable comments.

Second, the linear regression model applied to the relationship between COM position and MG activity characterize how these variables are related only on longer time-scales, giving the impression of a ‘contradictory’ relationship as the authors describe. Activation of the MG is associated with forward COP movement, which causes backward acceleration of the COM. Thus, examining the relationship between MG activity and horizontal COM acceleration would reveal a less contradictory relationship and help to more fully explain how these variables interact.

Third, and related to the above point, more sophisticated analyses of the relationship between the MG and COM movement could have illustrated how the COM started to move forward in the experimental data. Instead of simply looking at mean GC and TA activity in the time between CS and FS, the authors could have attempted to generate waveform averaged EMG aligned to the onset of forward COM movement. Presumably, this would have revealed an initial suppression of GC activity that permitted gravitational-induced forward COM shift.

Thank you very much for your valuable comments. Regarding the relationship between COM and GC, as described above, by examining the relationship between the magnitude of COM displacement and changes in GC activity, we were able to identify a consistent and non-contradictory relationship (Fig. 2G), as the reviewer pointed out, in which the instability induced by COM shift is compensated for by GC activity.

In addition, we found the reviewer’s comment regarding whether such a relationship could be observed in the temporal relationship between COM and COM acceleration to be particularly important. We also agree that the question of whether there exists a specific timing within the predictive period at which COM motion begins, and whether such timing can be identified, represents an important characteristic of the behavior. We sincerely appreciate the reviewer raising these points.

To address these questions, we derived COM velocity and acceleration and analyzed their relationships with GC activity and COM motion. First, the time series of COM position, velocity, and acceleration during the predictive period are shown in Fig. A below. Data from participant 1 are shown as an example, and similar patterns were observed in the other participants.

During the predictive period, COM motion exhibited large continuous oscillations in the anterior-posterior direction, as shown in Fig. A. In particular, COM acceleration showed oscillatory behavior centered around zero. In Fig. B, the horizontal axis represents COM position, velocity, or acceleration, and the vertical axis represents the corresponding level of GC activity. GC activity tended to increase when COM position and velocity were large; however, these relationships were not clearly defined. The relationship between COM acceleration and GC activity was even less apparent.

Because the predictive period in the present experiment precedes any change in the floor condition, the upright environment largely reflects the characteristics of quiet standing. Previous studies of quiet standing have reported that the trajectory of COM and COM velocity forms a limit cycle, and that control inputs are applied when COM position and COM velocity are in specific states. In intermittent control frameworks, this typically occurs when COM position and COM velocity are both positive or both negative. Based on this background, we plotted the trajectory of COM position versus COM velocity during the predictive period from cue onset to floor-tilt onset and represented GC activity at each state using color. This allowed us to visualize the timing at which GC activity was expressed. The resulting analysis has been newly added as Fig. 4.

Figure 4 shows that, during the early phase of the predictive period, the COM position-COM velocity trajectory forms a limit-cycle pattern similar to that observed during quiet standing. Subsequently,

while maintaining this cyclic structure, the COM gradually shifts forward. In addition, GC activity was found to increase particularly when the trajectory was located in the upper-right region of the cycle. This finding indicates that GC activity does not initiate at a specific time point, but rather emerges when the COM state transitions through the cycle and reaches the upper-right region. In other words, GC activity is modulated in relation to the phase of the COM cycle.

To examine whether a similar structure is observed in the simulation, we performed the same analysis and obtained the results shown in Fig. 6H. The simulation results revealed that the COM trajectory initially forms a cyclic pattern during the early phase of the predictive period and then progresses along a semicircular path as the COM moves forward. When the COM reaches an anterior position, GC activity increases in a manner that suppresses divergence of the COM velocity. This behavior closely resembles the trajectories and activity patterns observed in the experimental data.

The revisions made in response to the reviewer's comments can be summarized as follows.

We added **Figure 4**, illustrating the relationship between the COM position-COM velocity trajectory and GC activity.

In conjunction with this figure, we added the following explanation to the manuscript.

Line 199-209:

To further investigate the COM states in which GC is activated, we plotted the phase portrait of the COM^{38,39} (the relationship between COM position and COM velocity) during the cue period in Fig. 4, with GC activity at each state represented by color. As shown in the figure, the COM-COM velocity relationship forms a cyclic state pattern similar to that of quiet standing, and the COM exhibits a state transition in which it moves forward while preserving the cyclic shape. Furthermore, GC activity is particularly pronounced in the upper-right region of each cycle, corresponding to states in which the COM is located anteriorly and COM velocity is positive. This indicates that GC activity repeatedly switches its intensity depending on the state of the COM cycle. In other words, the COM limit-cycle dynamics and intermittent muscle activation previously reported during quiet standing^{38,40} are also suggested to be utilized during the predictive period.

In addition, we made corresponding changes in the simulation analysis.

We added **Fig. 6H** showing the relationship between the COM position-COM velocity trajectory and GC activity in the simulation.

The added explanation is as follows.

Line 260-270:

To investigate the relationship between COM shift and GC activity, we plotted the relationship between COM shift and changes in GC activity for each participant between CS and FS (Fig. 6G). The results revealed a relationship similar to that observed experimentally (Fig. 2G), in which greater COM displacement was associated with larger changes in GC activity (linear regression: $R^2 = 0.75$, $P < 0.001$). Furthermore, as in Fig. 4, we plotted the phase portrait of COM and represented GC activity using color, yielding the pattern shown in Fig. 6H. The trajectory initially exhibited a limit-cycle-like pattern during the early phase of the predictive period (left side), after which it shifted along a semicircular path. On the right side, where the COM was positioned anteriorly, GC activity increased, accompanied by a reduction in COM velocity. These results demonstrate trajectory and activity patterns similar to those observed in the experimental data (Fig. 4).

In Methods section, we described the method for obtaining the COM velocity as follows.

Line 582-587:

To construct trajectories of COM position and velocity during movement (COM phase portraits^{38, 39}), the COM velocity time series was derived by numerically differentiating the COM position time series. Prior to numerical differentiation, the COM position data were low-pass filtered at 5 Hz using a fourth-order Butterworth filter. Numerical differentiation was then performed using a five-point differentiation method.

We believe that these revisions adequately address the reviewer's comments by clarifying the interaction between COM motion and GC activity and the timing of GC activation. We sincerely thank the reviewer for these valuable comments.

2. While I laud the authors for combining their experimental data with a neuromechanical model, there are several results related to their model that make we question its validity.

(A) In the analyses where the authors adjust the weight of control inputs to examine how changes in the efficiency of the predictive controller changes the anticipatory lean, it doesn't look as though the magnitude or profile of the lean between CS and FS changes at all. Rather, it looks to stay the same, but the baseline target position changes, resulting in a larger absolute lean. This same observation applies to the GC activity. This invariant behavior is not discussed by the authors, but is instead characterized on the basis of the absolute forward position and GC activity.

Thank you very much for your comments. With regard to the magnitude and profile of the changes

between CS and FS, we note that in the experimental results these quantities were presented as slopes, whereas in the simulation results only absolute values were shown. For this reason, as the reviewer pointed out, it was difficult to directly compare the changes between CS and FS in the simulation with those observed experimentally. We appreciate the reviewer's helpful comment highlighting this issue.

To address this point, we evaluated the changes between CS and FS and added the results to the following figures. Specifically, the following revisions have been made:

- We added plots showing both the difference and the slope in Fig. 7A and B.
- We also added corresponding plots of the difference and slope in Fig. 6C, D, and F, enabling direct comparison between the simulation and experimental results.

Inspection of these newly added results (Fig. 7A and B, and Fig. 6C, D, and F) shows that GC activity changes between CS and FS in the simulation in a manner similar to that observed experimentally. Furthermore, to examine whether GC activity increases in accordance with COM motion, we also made following revision:

- We added a plot showing the relationship between forward COM displacement and the increase in GC activity (Fig. 6G).

As a result, consistent with the relationship observed in the experiment (Fig. 2G), the simulation results also revealed that GC activity increases as COM displacement increases (linear regression: $R^2 = 0.75$, $P < 0.001$). This finding indicates that the phenomenon observed experimentally, in which GC activity is generated to counteract the forward ankle torque associated with forward COM shift, is successfully reproduced in the simulation.

To incorporate these results, we added the following description to the manuscript:

Line 260-264:

To investigate the relationship between COM shift and GC activity, we plotted the relationship between COM shift and changes in GC activity for each participant between CS and FS (Fig. 6G). The results revealed a relationship similar to that observed experimentally (Fig. 2G), in which greater COM displacement was associated with larger changes in GC activity (linear regression: $R^2 = 0.75$, $P < 0.001$).

Through these additions and revisions, we believe that the relationship between forward COM shift and GC activity in the simulation is now more clearly demonstrated and directly comparable to the experimental results.

(B) Furthermore, in subsequent analyses where they manipulate the initial target position, such that the COM some times starts well behind the ankle joint (-50 mm), the magnitude of the preparatory lean between CS and FS doesn't change. This is somewhat baffling and is unlikely to be replicated by experimental data since a 20-30 mm lean starting a from a -50 mm position still leaves the individuals' COM behind the ankle joint and, therefore, in a very precarious position if they're expecting a backward platform tilt.

Thank you very much for your comments. We fully agree with the reviewer that, in predictive COM motion prior to the onset of floor tilt, shifting the COM anterior to the ankle is a strategy to maintain stability after the tilt. We also completely agree that, if the **initial target position** were changed, it would indeed be puzzling for the COM to remain posterior to the ankle even after the predictive movement.

However, the condition used in Fig. 8 corresponds to a **target position**, rather than an **initial target position**. That is, even after the onset of floor tilt, the target was set such that the COM would be located posterior to the ankle. As the reviewer noted, under this condition the COM remains posterior to the ankle after the tilt, but this outcome is a direct consequence of the condition imposed in the simulation. As the reviewer pointed out, this condition may be considered relatively demanding from an experimental perspective, as it would require instructing participants to stand while intentionally keeping their COM as close to the heel as possible. Nevertheless, although this is a challenging condition, we believe that it represents a feasible behavior, as shown in Fig. 8B, where stability is maintained through TA activity when the COM is located posteriorly.

The reason we changed a **target position**, rather than an initial target position, was to ensure that the system always started from a steady state. If the initial position and the target position differ, both goal-directed COM motion toward the target and predictive COM motion would be simultaneously present. In contrast, by keeping the initial and target positions identical, we can isolate and examine the effects of purely predictive COM motion. We acknowledge that our original description of the simulation conditions was insufficiently clear and may have led to misunderstanding.

To address this issue, we added the following sentence at **Line 308-311** to explicitly clarify the simulation conditions:

Here, in the simulation, the target position and the initial position were always set to be identical in order to reduce the influence of transient movements from the initial position to the target position

and to focus on movements based on prediction.

In addition, we revised the manuscript to consistently indicate that the condition corresponds to a target position.

(C) While the authors offer a possible explanation for the complete absence of TA activity and conspicuous increase in GC activity following platform tilts in their simulations, it does not inspire confidence in the other aspects of their model.

This is all to say that the validity of the neuromechanical model is dubious and the authors claim that the results from their neuromechanical match their experimental data is somewhat misleading.

Thank you very much for your comment. As you correctly pointed out, in the previous version of the manuscript we did not sufficiently discuss, based on solid evidence, the correspondence between the experimental data and the model with respect to the absence of tibialis anterior (TA) activity and the increase in gastrocnemius (GC) activity observed after floor tilt onset. As a result, the overall validity of the model was not sufficiently clear.

To address this issue, we revised the manuscript in two main ways: **(1)** we reexamined the experimental data to clarify how GC activity emerges after tilt onset, and **(2)** we extended the model to incorporate this factor and revised the discussion accordingly. The specific changes are described below.

- (1)** As noted in the reviewer's comment, GC activity during the tilt period exhibited a large, impulse-like response after tilt onset. We therefore analyzed the time from tilt onset to the peak of this activity and found that GC exhibited a prominent peak approximately 85 ms after tilt onset, corresponding to the early component of the long-latency response (LLR). Because the latency of this response did not differ between the Cue and No Cue conditions, we concluded that this activity has properties distinct from the predictive control targeted by the proposed model. Accordingly, reproducing this response requires an additional mechanism beyond predictive control.
- (2)** To examine whether the experimentally observed post-tilt activity could be reproduced by incorporating such a mechanism, we performed simulations using an extended model in which a preprogrammed, tilt-triggered activity corresponding to the LLR was added to the proposed model. With this extension, impulse-like GC activity comparable to that observed experimentally was reproduced, followed by the emergence of TA activity.

These results are now presented in a new subsection entitled **Extension of the model to reproduce reflexive activity during floor tilt** in the Results section, together with newly added simulation results (Fig. 9), as described below (Lines 344-371):

The proposed model was designed to reproduce the forward COM shift and GC activity during the predictive period preceding floor tilt, and showed that these behaviors can be reproduced by model predictive control. However, when muscle activity during the floor tilt itself was examined, differences between the experimental and simulation results became evident. In the experiments, GC exhibited a transient, impulse-like activation following floor tilt onset, followed by TA activation (Fig. 2E; similar responses were observed in other participants, Supplementary Fig. S3). In contrast, the simulation showed a gradual increase in GC activity during the same period, and no TA activation was observed (Fig. 6E).

To characterize the experimentally observed impulse-like GC activity, we quantified the timing of GC activation during the tilt period. The first peak of GC activity occurred at 83.0 ± 52.6 ms in the no-cue condition and 85.0 ± 50.7 ms in the cue condition after tilt onset (see Supplementary Table S5 for peak latencies of individual subjects). These latencies correspond to the early component of the long-latency response (LLR)¹². No significant effect of cue condition on peak latency was observed (two-way ANOVA with factors Subjects and Cue condition: Subjects, $F = 1.17$, $df = 9$, $P = 0.31$; Cue condition, $F = 0.161$, $df = 1$, $P = 0.69$).

Based on these observations, we extended the control input by adding muscle activity corresponding to the LLR, in addition to the existing model predictive control-based input. Specifically, GC activation was applied during a 100-ms window following floor tilt onset. With this extension, the simulation reproduced an impulse-like activation of the GC after tilt onset, followed by TA activity (Fig. 9E). Consistent with the experimental results (Fig. 2A), the COM motion decreased during the tilt phase (Fig. 9A), and a rapid decrease in hip angle following tilt onset was also reproduced (Fig. 9B). The transient increase in COM observed after tilt termination in Fig. 9A was not present in Fig. 2A (Subject 7), but similar behavior was observed in other participants (Subjects 1, 2, 4, and 9; Supplementary Fig. S1).

Based on these additional results, we also revised the Discussion section as follows (Lines 461-483):

While the model predictive control-based model used in this study successfully reproduced the COM shift and GC activity observed prior to floor tilt, it failed to account for the impulse-like GC activation during the tilt itself based on MPC alone. To further examine the impulse-like GC activity observed after tilt onset, we analyzed its temporal characteristics and found that it differed from the control properties observed during the predictive period. Specifically, whereas the forward COM shift and GC activity during the predictive period varied depending on the presence or absence of

the cue, no significant difference was observed in the latency of GC activation after tilt onset between the Cue and No Cue conditions. This result suggests that muscle activity following tilt onset may be generated by mechanisms distinct from the cue-dependent predictive control operating during the predictive period. Indeed, in upper-limb movements, long-latency responses (LLRs) have been modeled as feedback processes incorporating prediction⁵⁹, whereas in upright postural control, LLRs have been reported to behave as preprogrammed signals triggered by muscle stretch⁶⁰, with the early component in particular exhibiting preprogrammed characteristics⁶¹. Based on these findings, we extended our model by incorporating an additional term representing reflexive activity following floor tilt, in addition to the MPC responsible for predictive control. With this extension, the impulse-like GC activation during the floor tilt and the emergence of TA activity were reproduced, while the overall postural behavior remained stable under the combined action of predictive control and reflexive input (Fig. 9). Although further discussion is required regarding how LLRs during floor tilt should be modeled, the present results demonstrate that the proposed postural control model based on MPC can be extended to describe not only anticipatory adjustments but also responses during the perturbation itself.

In addition, the Methods section has been revised as follows (**Lines 660-665**):

To reproduce muscle activity during floor tilt, additional simulations were performed using an extended model in which a term representing reflexive activity was added to the control input based on MPC. Specifically, a constant activation (50%) was applied to the GC during a 100-ms window starting 100 ms after floor tilt onset. To display the GC results, the simulated GC activity was filtered using the same 10-Hz low-pass filter (fourth-order Butterworth filter) as applied to the experimental data.

Finally, we note that the primary focus of this study was to model the control principle underlying the forward shift of the COM during the predictive period using predictive control, and reproducing muscle activity during the tilt itself was not the main objective. In particular, further investigation is required to determine an appropriate modeling framework for long-latency responses following tilt onset. Nevertheless, the present model extension demonstrates that muscle activity during the tilt period can be reproduced to a certain extent, thereby clarifying both the applicability and the limitations of the proposed model. We thank the reviewer for this insightful comment, which helped us to improve the clarity and rigor of the manuscript.

Minor issues (no order):

- Were only backward tilt perturbations applied? If so, why was this done? Presumably participants would have adopted a slight pitched forward position knowing that the perturbations could only come in one direction, which may have blunted additional anticipatory leaning following the CS (as was seen in some subjects).

Thank you very much for your insightful comment. In the present study, only backward tilt perturbations were applied. This choice was based on the experimental design of Kolb et al. (2002), and was made in order to clearly extract and analyze muscle activity during the predictive period preceding floor tilt, as well as to explain the generation of this activity using a mathematical model.

In studies of postural control, the activity of the plantarflexor muscle group, particularly the gastrocnemius (GC), has been widely used as a primary indicator of postural reflexes and predictive activity in response to floor tilt perturbations (Nashner, 1976; Horak & Macpherson, 1996; Kolb et al., 2002). Following these previous studies, we therefore selected backward tilt perturbations as a condition that allows stable evaluation of predictive muscle activity.

As pointed out by the reviewer, it cannot be excluded that applying perturbations in only one direction influenced the participants' posture. In the present study, participants were informed only that perturbations would occur, and the direction of the perturbation was not specified. Therefore, if participants had learned that the perturbation direction was unidirectional and systematically changed their posture, such effects would be expected to appear most prominently in the early trials. However, as shown in Fig. 3, no clear systematic differences in muscle activity were observed between the early trials (trials 1–5) and the later trials. This result suggests that the influence of unidirectional perturbations on posture and predictive muscle activity was not substantial, at least within the range observed in the present study.

Based on these considerations, although the influence of unidirectional perturbations cannot be completely excluded, we believe that its effect was limited. We have added this discussion to the Discussion section as below (Line 484-496):

In the present study, based on the experimental design of Kolb *et al.*^{9, 62}, only backward tilt perturbations were applied to investigate predictive muscle activity prior to floor tilt. In studies of postural control, the activity of the plantarflexor muscle group, particularly the GC, has been consistently used as a primary indicator of postural reflexes and predictive activity in response to floor tilt perturbations^{12, 63}. On the other hand, it cannot be excluded that restricting perturbations to

a single direction influenced the participants' postural configuration. In particular, if participants came to expect only backward tilts, they might have adopted a slightly forward-leaning posture compared with the initial phase of the experiment, when the perturbation direction was not yet anticipated. However, as shown in Fig. 3, no clear systematic differences were observed between the early trials (trials 1–5) and the later trials. This result suggests that the influence of unidirectional perturbations on posture and predictive muscle activity was limited within the range observed in the present study.

- In the introduction when discussing automatic postural responses (lines 44-48), the direction of the translational disturbance should be made clear.

Thank you for this helpful comment. We have revised the Introduction to clarify the direction of the translational disturbance when discussing automatic postural responses. Specifically, we now explicitly state that the translational disturbance is “directed forward in the anterior–posterior direction” (Line 44). In addition, for tilting disturbances, we clarified that “a tilting disturbance in the toe-up direction is applied, which induces ankle dorsiflexion and results in a backward perturbation of the body” (Lines 47-48). These revisions have been made in Lines 44-48 of the revised manuscript.

Response to Reviewer #2

I'd like to thank the authors for carefully considering my criticisms of their previous draft. They've made considerable changes to the paper, including the addition of several new figures/analyses that I think have greatly improved their manuscript. I have only two minor suggestions for them to consider:

Thank you very much for your careful and constructive review.

The authors use a bit of a strawman argument to justify their study. At points in the abstract, introduction, and discussion, it is described as though they want to know if the anticipatory (predictive) postural adjustment prior to predictable perturbations are achieved through predictive rather than feedback control mechanisms. This is self-evident as these adjustments are predictive in their nature and therefore cannot be produced through feedback control. I believe this rationale should be modified to emphasize the optimal control aspect of their work rather than contrasting predictive and feedback control.

Thank you very much for this helpful suggestion. We agree that the previous version of the manuscript may have unintentionally framed the study as a contrast between predictive and feedback control. Our intention, however, was not to argue against feedback control, but rather to clarify the computational principles underlying anticipatory control inputs.

In response to the reviewer's comment, we revised the manuscript to emphasize the role of predictive optimal control rather than contrasting predictive and feedback mechanisms.

Specifically, we revised the following sections:

- **Abstract (Lines 17-19):**
“The control mechanism underlying this coordination between forward COM motion and GC activation remains unclear. Here we investigate whether such behavior can be explained within a predictive optimal control framework.”
- **Introduction (Lines 73-76):**
“Because anticipatory COM movements occur before the disturbance, their generation requires predictive control mechanisms. However, the computational principles that generate such predictive control inputs remain unclear.”

- **Discussion (Lines 386-388):**

“To explain such active movements before environmental changes, we hypothesized that these anticipatory COM movements arise from predictive optimization of future states based on internal models.”

- **Discussion (Lines 402-403):**

“These results suggest that the observed predictive behavior can be explained within the framework of model predictive control.”

- **Discussion (Lines 527-529):**

“This has been difficult to address using conventional posture control models, but can be more naturally examined within the MPC framework used in the present study.”

These revisions clarify that the goal of the study is not to contrast predictive and feedback control, but rather to investigate how predictive optimization of future states can generate the observed anticipatory COM movements and associated muscle activity.

Some discussion of what the center of pressure (COP) is doing during the anticipatory postural adjustments would be nice. The authors discuss the COM-COM velocity phase plane analyses (which I appreciated that they added to the paper) and note how the GC activity is generally greatest when the COM is in a forward position while moving forward. They then suggest that mechanisms other than muscle-generated torques are contributing to the preparatory forward COM movement (lines 199-216). Here, the authors make a statement which makes it seem as though this behavior is somewhat unexpected: “However, when focusing on the fact that GC can generate only posterior ankle torque while the COM is being shifted forward, the direction of the generated torque is opposite to the direction of the intended COM shift. This inversion between the direction of control and the direction of muscle-generated torque suggests that forces other than muscle-generated torque contribute to moving the COM forward.”

What’s happening during these points in time is the increased GC activity is causing the COP to move forward ahead of the COM to slow its acceleration. This is why the GC is highly active while the COM is still moving forward. This is all part of what has been referred to previously as a ‘drop and catch’ pattern that occurs repeatedly during quiet standing during forward sways (see work by Loram and colleagues between 2001-2006). I.e., first, the GC relaxes, this causes the COP to fall behind the COG and allows the COM to accelerate forward (the drop). As the COM moves forward, the GC then turns on to slow the COM’s acceleration and stop it in its new position (the catch). Adding a small statement of these COP-COM dynamics would be helpful to clarify what’s happening during their forward sways.

Thank you for this insightful comment and for pointing out the importance of considering COP-COM dynamics when interpreting the relationship between COM motion and gastrocnemius (GC) activity.

Following the reviewer's suggestion, we have added an explanation of the COP-COM dynamics associated with forward sway during standing. In particular, we now discuss the "drop-and-catch" mechanism described in previous studies (Loram et al., 2001; Loram & Lakie, 2002; Loram et al., 2005), in which relaxation of the plantarflexor muscles allows the COM to accelerate forward while the center of pressure (COP) moves posterior to the COM, followed by plantarflexor activation that shifts the COP anteriorly and decelerates the COM. This explanation clarifies how GC activity can increase while the COM is still moving forward.

This discussion has been added to the Results section (**Lines 218-226**):

- "A similar relationship between the direction of COM motion and ankle torque has been observed in previous studies of quiet standing^{41, 42} and has been interpreted in terms of the relationship between the COM and the center of pressure (COP)³⁸. These studies have shown that forward sway during quiet standing can arise through a "drop-and-catch" mechanism^{42, 43}. In this mechanism, relaxation of the plantarflexor muscles causes the COP to move posterior to the COM, resulting in forward acceleration of the COM (drop phase). As the COM moves forward, the plantarflexors are subsequently activated, shifting the COP anterior to the COM. This reduces the forward motion of the COM and stabilizes it at a new position (catch phase)."

In addition, we further clarified that gravity-generated torque together with GC activity can contribute to the forward shift of the COM while maintaining stability.

We believe that this addition helps clarify the biomechanical interpretation of the observed COM-GC relationship and better connects our results to the existing literature on postural control.